# Investigation of Thundercloud Features in Different Regions

Andrei Sin'kevich [1,*], Bruce Boe [2], Sunil Pawar [3], Jing Yang [4], Ali Abshaev [5], Yulia Dovgaluk [1], Julduz Gekkieva [6], Venkatachalam Gopalakrishnan [3], Alexander Kurov [1], Yurii Mikhailovskii [1], Marina Toropova [1] and Nikolai Veremei [1]

1 Main Geophysical Observatory, Department of Geophysical Monitoring and Investigation, 194021 St. Petersburg, Russia; dovgaluk35@mail.ru (Y.D.); remotesensing@mail.ru (A.K.); yupalych@yandex.ru (Y.M.); marina-toropova@mail.ru (M.T.); veremey@gmail.com (N.V.)

2 Weather Modification International, Fargo, ND 58102, USA; bboe@weathermodification.com

3 Department of Thunderstorm Dynamics, Indian Institute of Tropical Meteorology, Pune 411008, India; pawar@tropmet.res.in (S.P.); gopal@tropmet.res.in (V.G.)

4 Institute of Atmospheric Physics, Chinese Academy of Sciences, Beijing 100029, China; yangj@mail.iap.ac.cn

5 Hail Suppression Research Center "ANTIGRAD", 360004 Nalchik, Russia; abshaev.ali@mail.ru

6 Department of Cloud Seeding, High-Mountain Geophysical Institute, 360030 Nalchik, Russia; julduz_gekkieva@mail.ru

* Correspondence: sinkevich51@mail.ru; Tel.: +7-812-297-43-90

**Abstract:** A comparison of thundercloud characteristics in different regions of the world was conducted. The clouds studied developed in India, China and in two regions of Russia. Several field projects were discussed. Cloud characteristics were measured by weather radars, the SEVERI instrument installed on board of the Meteosat satellite, and lightning detection systems. The statistical characteristics of the clouds were tabulated from radar scans and correlated with lightning observations. Thunderclouds in India differ significantly from those observed in other regions. The relationships among lightning strike frequency, supercooled cloud volume, and precipitation intensity were analyzed. In most cases, high correlation was observed between lightning strike frequency and supercooled volume.

**Keywords:** thundercloud; radar; cumulonimbus; electrical strike; lightning; Meteosat

## 1. Introduction

The development of powerful cumulonimbus often leads to dangerous phenomena: heavy rainfall, squalls, tornadoes, hail, and lightning. The formation of these phenomena is closely related to the microphysical, dynamic, and electrical characteristics of clouds. The current understanding of cloud and precipitation formation should also include the role of electric fields and discharges on various scales in clouds, from corona to lightning [1–5].

Currently, most scientists consider the interaction among ice particles as the main mechanism of charge generation [6]. Taking into account the significance of the contact mechanism of electrification for charging cloud particles, a close correlation might be expected between the microphysical parameters of the cloud and the frequency of electric discharges. Correlation between the lightning flash rate and the cloud top height [7,8] were attempted, as well as with the maximum radar reflectivities observed at different heights [9–12], and also the number of ice crystals, estimated from radar data [13–16]. It is demonstrated in [17] that there are rather close relationships between the frequency of lightning and the total mass of particles in the upper part of the cloud, and also the mass of graupel and hailstones. These relationships are manifested in both supercell and multicell clouds. Other radar parameters were also examined, notably those which depend on: the number and size of ice particles in the cloud; the characteristics of precipitation in the form of ice particles [2,11,16]; the total mass of particles in the upper part of the cloud; the mass of graupel and hailstones [17]; the volume of the cloud containing mixed-phase

hydrometeors [18]; among other characteristics [19–22]. Synchronous measurements using the radar and LIS instrument installed at the TRMM satellite were carried out during the period 1998–2010. These demonstrated that correlation between the cloud top height and the lightning frequency is insignificant but the lightning frequency depends more on the radar volume of the cloud that contains mixed-phase hydrometeors [18].

A significant correlation was obtained between the frequency of lightning flashes and ice-phase precipitation [2,11,14,15,23,24]. The rainfall volume per cloud-to-ground lightning flash was found to be from $0.7 \times 10^4$ to $6.4 \times 10^4$ m$^3$ [24]. In most cases, discharges preceded precipitation. All researchers note the regional and seasonal features of the revealed relationships.

It has been shown that the formation of the crystal fraction in convective clouds is accompanied by the increase in electric field strength [25,26], which is highly correlated with the integral characteristics of the supercooled part of the cloud defined by the number of rough ice particles in the cloud [27]. The importance of corona discharges in the process of cloud charge generation remains poorly explored, although there are hypotheses and experimental data which suggest that the contribution of these processes is significant [3].

Field experiments have demonstrated that the frequency of electric discharges in the cloud is proportional to the updraft speed to the sixth power [19], while the authors of [21,28] found that the lightning frequency was proportional to the updraft speed to the power of 4.55.

The presented brief review identifies Cb characteristics typical to the thundercloud-transition stage of Cu development, important for understanding cloud electrification physics. This also has practical implications for lightning prediction.

Attempts to establish relationships between lightning and the dynamic and micro-physical cloud characteristics have primarily been made through remote sensing in recent years. Lightning characteristics, including lightning frequency, depend on Cb properties. It follows from the literature review that the crystal phase likely plays a role in lightning formation. Those specific habits of crystal phase which are most important for Cu electrification remain a challenge, and remain a topic for future investigations. At the same time, the varied conditions under which thunderclouds develop suggest that Cu electrification and hence lightning formation might result from differing mechanisms. This statement is supported by the regional variation in Cb properties when lightning is first observed.

The objectives of the present paper were to retrieve thundercloud characteristics and to quantify the interrelations between the parameters of lightning strikes and Cu characteristics during lightning events in different parts of the world. It is postulated that thundercloud properties differ significantly according to the varying climates under study: north and south Russia, India and China. Hence, one can expect that relations between lightning characteristics and Cu characteristics will also be different, suggesting that differences in Cu electrification play an important role in lightning initiation frequency. Special attention is paid to lightning formation in a supercell Cb, which developed in the north Caucasus of Russia.

The following data illustrate a significant difference between thundercloud characteristics in Russia and China, in comparison with India. Moreover, thundercloud characteristics also differ for northern and southern Russian regions. These features can be explained by differing troposphere depth, the height of a zero-degree isotherm, and other troposphere properties: temperature, humidity, aerosol content. The observations confirm the importance of a crystal phase in Cu electrification. A volume with strong radar reflectivity above the zero-degree isotherm, a prerequisite for lightning formation, is assessed.

This paper is organized as follows: the field projects, characteristics of the radars, lightning detection systems, and methods are described in Section 2. Cloud characteristics before lightning formation and at the time of first lightning are discussed in Section 3.1. Characteristics of thunderclouds during the periods of low frequency (LF) lightning flashes are given in Section 3.2, and similarly, the characteristics for very high frequency (VHF) strikes are found in Section 3.3. Relationships between LF flashes, supercooled cloud

volume, and maximum precipitation intensity are discussed in Section 3.4, and the analogous data for VHF are discussed in Section 3.5. Information about lightning current and transferred charges is presented in Section 3.6. Discussion and conclusions are in Sections 4 and 5.

## 2. Materials and Methods

### 2.1. Observations

The field projects discussed herein include studies of Cu and thunderclouds in Russia (in the vicinity of St. Petersburg and in the north Caucasus near Pyatigorsk), in India (in the states of Maharashtra and Karnataka), and in China (in the vicinity of Beijing)—as shown in Figure 1.

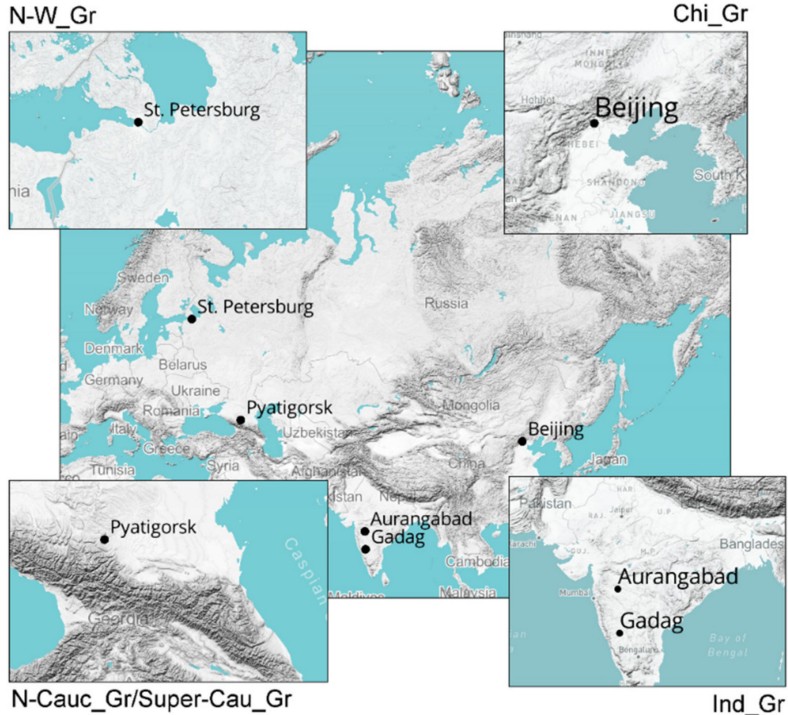

**Figure 1.** Regions under study: N–W_Gr—St. Petersburg (Russia); N-Cauc_Gr/Super-Cau_Gr— north Caucasus (Russia); Ind_Gr—the states of Karnataka and Maharashtra (India); and Chi_Gr— Beijing (China).

The development of convective processes and thunderstorm activity are governed by the geography and climate of regions under study. St. Petersburg is located in the northwestern part of the East European Plain. The climate is moderate, transitional from continental to maritime. The climate of St. Petersburg is strongly influenced by two large waterbodies: the Gulf of Finland (Baltic Sea) and Ladoga Lake. Weather processes are mainly determined by westerly disturbances and cyclonic activity—as can be seen in Figure 1 (NW_Gr).

Pyatigorsk is located in the foothills of the Caucasus Mountains. Its climate is defined as temperate continental. The mountains on the southern side protect the city from the influx of warm air from the Transcaucasia, while cold Arctic air can penetrate from the northern side. The city is located between the Black and Caspian Seas, but their influence is greatly weakened by the mountains, as can be seen in Figure 1 (N-Cauc_Gr/Super-Cau_Gr).

The areas of study in India are in relatively flat regions and are in sharp contrast to the mountainous areas in the north of the country (Tibetan plateau region). The climate is characterized as subtropical, as shown in Figure 1 (Ind_Gr).

Beijing lies in the northern part of the Great Plain of China and is protected by mountains on the north and west sides. The climate is humid subtropical or humid continental, as shown in Figure 1 (Chi_Gr).

In the regions of India and China under study, there is a strong influence from the Asian monsoon, which determines seasonal changes in the weather regime, mainly those of the influx of humid air and an increase in precipitation.

Detailed information about projects where data were obtained and summaries of the Cu characteristics observed in each case were presented in earlier publications [1,29–39]. All lightning flashes, which were detected in a cloud during a radar scan, were counted and the average frequency for a scan in $min^{-1}$ was determined. This frequency was used for further data processing.

There are two types of data which were analyzed here. The first one is used to compare the radar characteristics of different clouds at the moment of the beginning of lightning activity to those prior to lightning. Results of this investigation are presented in Section 3.1.

The statistical radar characteristics of Cu transitioning to thunderclouds in three regions—India, the north Caucasus (Russia), and northwestern Russia—are discussed [38]. Data for India were obtained during the period from late September to early October 2019. Radar data in the north Caucasus were acquired between May and September during the period 2010–2018, and in northwestern Russia, these were obtained between May and September during the period 2017–2018. Clouds typical of the end of the monsoon period were observed in India, and continental clouds of the warm season for Russia.

About 60 convective clouds transitioning to thunderclouds were selected for each region. Radar characteristics were tabulated for them 10 min before the beginning of lightning activity, and during the first discharges (LF and/or VHF flashes). Consequent radar scans were under consideration: the first when lightning was observed and the second 10 min prior to lightning.

The second type of data is analyzed in Sections 3.2–3.6. These data were retrieved from detailed case studies of thunderclouds, observed in Russian regions and China. Temporal variations of Cu characteristics were analyzed earlier for each case study. Attempts to understand the physical processes leading to lightning were made. To obtain the needed information radar, satellite, rawinsonde, and lightning detection network measurements were used alongside numerical modeling, including 3D modeling. These data were also used in the current investigation.

This analysis compares the Cu characteristics of the four regions at the time, when the clouds were already transformed into thunderclouds. Radar characteristics of clouds were recorded when lightning was observed and then tabulated. Very thorough analysis of cloud evolution was carried out earlier, as each lightning stroke observed was matched with the position of a studied Cu.

Brief reviews of each cloud group studied are given below.

The first group includes four Cb which were studied in northwestern Russia in the vicinity of St. Petersburg (lat/lon ~60°/30°). These were Cb which developed during warm periods in the vicinity of the city: thunderstorm with hail, 22 July 2017 [33,36]; thunderstorm with waterspout over the Gulf of Finland, 12 August 2018 [30,35]; thunderstorm with waterspout over Ladoga Lake, 6 August 2018 [29]; thunderstorm with hail near Petrokrepost, 25 May 2017. In total, 23 radar scans at the moment of LF flashes were under consideration. The collective characteristics of the clouds are summarized, and henceforth referred to as N–W_Gr.

The second group includes two clouds studied in the vicinity of Pyatigorsk, Russia (north Caucasus) (lat/lon ~44°/43°). Both clouds were thunderstorms with hail. The first was observed on 29 May 2012 [31] and the second was observed on 12 June 2015 [37]. A total of 71 radar scans at the moment of LF flashes and 80 radar scans at the moment of VHF flashes are under consideration. This group is referred to as N-Cauc_Gr.

The third group consists of one supercell Cb observed in Russia, which originated near the Black Sea, and crossed Caucasus on its way to the Caspian Sea (19 August 2015). It

was a deep and long-lived Cb with hail and tornado [34]. It was analyzed separately from N-Cauc_Gr as it was a rather unique event, and the characteristics of this cloud differed significantly from "typical Cb" in that region. In total, 239 radar scans at the moment of LF flashes and 200 radar scans at the moment of VHF flashes are under consideration. This cloud is referred as Super-Cau_Gr.

The fourth group is the Cb observed in the State of Maharashtra, India, near Aurangabad (lat/lon ~20°/75°) on 4 October 2015 [39] and in the State of Karnataka, India, near Gadag (lat/lon ~15°/76°) on 15 September 2019. Visual observations and further data analysis showed that the occurrence of lightning in clouds in India was low. To aggregate the data sufficient for statistical analysis, all clouds in which lightning was observed were grouped. Twenty-nine thunderclouds, which were located within the range of radar, were studied. In total, 38 radar scans at the moment of LF flashes and 50 radar scans at the moment of VHF flashes are under consideration. These clouds were henceforth referred to as Ind_Gr.

The fifth group consists of one Cb which was observed on 7 August 2015 in the vicinity of Beijing, China (lat/lon ~39°/116°) [1]. A total of 9 radar scans at the moment of LF flashes and 10 radar scans at the moment of VHF flashes are under consideration. This thunderstorm is henceforth referred to as Chi_Gr.

The comparison of the Cu characteristics was based on the available data and includes different numbers of clouds and observations; hence it does not pretend to be a true climatological study, but the data processing was done in the same manner, and the results show trends in Cu characteristics and their differences depending on the geographical region.

## 2.2. Radars and Lightning Detection Networks

Data for India were obtained by C-band WR-100 radars sited in Aurangabad and Gadag. Radar data in the north Caucasus were acquired with a dual-wavelength S/X MRL-5 meteorological radar, located near Stavropol. An S-band Doppler weather radar was used in China. In northwestern Russia, the data were obtained with a dual-polarization Doppler DMRL-C C-band radar, located near St. Petersburg. Each radar continuously makes 360-degree scans of the atmosphere at different elevation angles. The main parameters of the radars are given in Table 1.

**Table 1.** Main parameters of the radars.

| Parameter | Radar | | | |
|---|---|---|---|---|
| | **WR-100** | **MRL-5, Channel II** | **DMRL-C** | **S-Band Radar** |
| Wavelength, cm | 5.4 | 10 | 5.3–5.5 | 10 |
| Scan period, min | 7.5 | 3.5 | 10 | 6 |
| Half power beam width, deg | 1.65 | 1.5 | 1.0 | 0.94 |
| Sensitivity, dB/W | - | −42 | −142 | −110 |
| Maximum detection range of meteorological objects, km | 100 (at a reflectivity of 0 dBZ) | 250 | 250 | 230 |

Errors in the radar measurements of Cu characteristics arose from instrumental errors and from the assumptions about single and incoherent scattering, hydrometeor shapes, dielectric properties, and the size distribution [40]. The differences in Cu characteristics in the regions under study may be to some extent due to the differences in the radar parameters, as shown in Table 1.

Lightning detection systems now exist around the world. These systems mostly sense radiation from electric discharges in two bands: low frequency (LF) and very high frequency (VHF). It is assumed that approximately 30% of cloud-to-cloud and cloud-to-ground discharges are registered in the LF range, and all discharges, including the cloud-to-cloud and intra-cloud, are registered in the VHF range [41].

Lightning locations from multiple systems were used, including the lightning location network (LLN), which uses 4 Vaisala LS8000 sensors in the north Caucasus region of Russia [31]; the Blitzortung network in northwestern Russia [33]; Beijing Lightning Network (BLNET) in China [42]; Lightning Information and Prediction System (LIPS) in India (https://www.tropmet.res.in/~lip/annual-reports/AR-English-2013-14.pdf; accessed on 9 August 2021).

All four networks detect the electromagnetic pulses of cloud-to-ground (CG) and intra-cloud (IC) discharges, but use different frequencies. LS8000 sensors work at two bands: 1–350 kHz and 110–118 MHz. Blitzortung works at frequencies from 3 to 30 kHz. The BLNET consists of 16 stations distributed over Beijing and the adjacent region, each of which has three kinds of sensors: 1.5 kHz–2 MHz (fast antenna), 10 Hz–1 MHz (slow antenna), and 69–75 MHz (very high frequency sensor) to detect lightning electro-magnetic signals at broad frequency bands. The waveform features of lightning pulses in each station were analyzed, and the information on the lightning type and arrival time recorded at each station was transferred to the central station for the three-dimensional lightning mapping. The detection efficiency and location accuracy for total lightning was 93.2% and 250 m, respectively [43]. LIPS sensors work at 1 kHz–12 MHz. LS8000, BLNET and LIPS sensors allow the differentiation of CG and IC discharges, and calculate their location, current, polarity, and various other time characteristics.

Among the LLNs presented, Blitzortung, being a non-commercial community-based worldwide LLN, has the most coverage. LIPS covers the entire territory of India, while BLNET and LS8000 only cover specific regions (Beijing and adjacent areas (China) and north Caucasian/Southern Federal Districts (Russia), respectively).

### 2.3. Thundercloud Criteria Y

Currently, the indirect thundercloud criteria $Y$, based on radar data, is used in Russia for assessing the lightning potential of storms [32,44]. It was developed on a statistically significant number of radar-derived parameters and is used for predicting thunderstorm probability:

$$Y = H \log Z_3 \tag{1}$$

where $H$ is the cloud top height (km); and $Z_3$ (mm$^6$/m$^3$) is the radar reflectivity at a height of 2–2.5 km above the zero-degree isotherm. In each case, the nearest rawisonde sounding was used to determine a height of the isotherm. The thunderstorm risk is defined by the relationship between the value of $Y$ and parameter $Y_{cr} = H_{-22} (\log Z_3)_{min}$, where $H_{-22}$ is the height of the $-22\,°C$ isotherm, $(\log Z_3)_{min} = 1.5$ is the minimum value of $\log Z_3$ in thunderstorms [32]. It was empirically found that the possibility of lightning appearance in the cloud is >90% if $Y > Y_{cr} + 14$; it is between 70 and 90% if $Y > Y_{cr} + 6$; and less than 70% if $Y > Y_{cr}$. The parameter $Y$ was used to further characterize the electrical state of Cu in this investigation. It can be concluded from the above that an increase in the $Y$ parameter means an increase in cloud electrification and the probability of lightning.

### 2.4. Thundercloud Reflectivity and Supercooled Volume

For a mixed-phase cloud, the radar reflectivity $Z$ (mm$^6$/m$^3$) is a sum of reflectivity due to drops and ice particles:

$$Z = Z_w + Z_i \tag{2}$$

For water drops, the reflectivity depends on their sizes to the sixth power for the Rayleigh-scattering approximation [45]:

$$Z_w = \sum_b N_b D_{bw}^6 \tag{3}$$

where $N_b$ is the drop number concentration and $D_{bw}^6$ is the drop diameter in each bin (size).

A radar reflectivity for ice crystal clouds can be found using the following equation [45]:

$$Z_i = 0.267 \sum_b N_b D_{bi}^6 \rho_i^2 \tag{4}$$

where $D_{bi} = (D_x + D_y)/2$, $D_x$, $D_y$ are the crystal sizes in perpendicular directions. Usually, some averaged crystal size is used. The $\rho_i^2$ is the square of the ice hydrometeor density. This is usually unknown, so approximation formulas from experimentation can be used [46]:

$$Z_i = 0.003 \sum_b N_b D_{bi}^{3.8}, \text{ for } D > 0.1\text{mm} \tag{5}$$

$$Z_i = 0.2 \sum_b N_b D_{bi}^6, \text{ for } D < 0.1\text{mm} \tag{6}$$

These formulas show that radar reflectivity is to a large extent determined by the presence of big particles. It depends on the size of ice particles to the 6th or 4th power (depending upon the (ice) density). Laboratory and numerous field experiments have shown that cloud electrification strongly depends on the presence of big ice particles, both graupel and ice crystals [6].

Hence, one can expect that cloud electrification and the formation of electrical discharges should strongly depend on the value of reflectivity and volume of the cloud containing large ice particles. These precepts were the motivation to study the relation between lightning frequency and cloud volumes with different (but strong) reflectivities. Previous investigations have shown that the greatest correlation with lightning frequency has been observed for the cloud volumes with reflectivity greater than 35 dBZ, though in some cases this was from 40 to 50 dBZ [29,30,33].

## 3. Results

### 3.1. Thundercloud Characteristics at First Lightning Strike

The following radar characteristics were analyzed: maximum height of the 5 dBZ reflectivity $H_{5\text{dBZ}}$ (km); the maximum reflectivity $Z_m$ (dBZ); the $Z_m$ height $H_{Zm}$ (km); and volumes dV35 and dV45 above the zero-degree isotherm with the reflectivity higher than reflectivity thresholds of 35 and 45 dBZ. Radar data for all the regions were horizontally and vertically averaged in 1 km steps.

The largest median in the lightning frequency distribution was noted in the North Caucasus (1.2 min$^{-1}$); it was 0.3 min$^{-1}$ in India and 0.2 min$^{-1}$ in northwestern Russia. These differences may be due to the regional properties of the clouds, differences in characteristics of lightning detection networks, and the measured types of discharges.

The radar characteristics of each regional cloud set were tabulated. The differences between the samples before and at the time of the first lightning discharges were analyzed. The distributions were not normal; therefore, the Wilcoxon signed rank test was used for their comparison. Using this method, the $p$-levels of significance ($p$-value) of the similarity of the medians of the characteristics under study in the clouds with and without discharges were calculated. A higher $p$-value means a smaller difference between the distributions of two samples, so this was used to assess the similarity of the analyzed distributions. The calculated $p$-values show that the transition of clouds to cumulonimbus in India was not accompanied by noticeable changes in the radar characteristics ($p > 0.1$), except for $H_{5\text{dBZ}}$ ($p = 0.01$), as shown in Figure 2b. In the north Caucasus and northwestern Russia, statistically significant differences in cloud characteristics were observed before the beginning of thunderstorm activity and during the first discharges, except for $H_{Zm}$ in the north Caucasus, as shown in Figure 2a–d. Figure 2 shows the distribution of the characteristics under study.

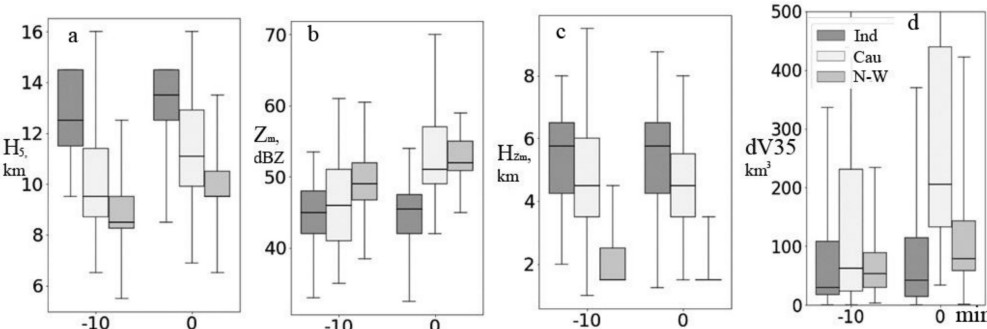

**Figure 2.** Box-and-whisker diagram of radar characteristics: (**a**) height of the 5 dBZ reflectivity $H_5$ (km); (**b**) the maximum reflectivity $Z_m$ (dBZ); (**c**) the $Z_m$ height $H_{Zm}$ (km); (**d**) volume dV35 above the zero-degree isotherm with the reflectivity higher than reflectivity threshold of 35 dBZ. The abscissa axis: two radar scans at the time of a lightning discharge (0 min) and 10 min before the discharge (−10 min). A box shows the 25 % and 75% quartiles, the line inside a box is the median, whiskers are the minimum and maximum in a sample. Regions (small box in upper right corner): India (Ind); north Caucasus (Cau); northwestern Russia (N-W).

The maximum height of the 5 dBZ reflectivity $H_{5dBZ}$ (km) is the only characteristic of clouds which increases during their transition to the thundercloud stage in all regions. It is the highest (the median is 13.5 km) in India, and the smallest (the median is 9.5 km) in northwestern Russia, which corresponds to a general increase in the depth of clouds with a decrease in latitude.

The maximum reflectivity of clouds transitioning into the thundercloud stage is mainly observed at altitudes of ±2 km relative to the zero-degree isotherm in all the regions, usually higher than the melting level in the north Caucasus and India, but lower in northwestern Russia. In the Russian regions, the altitude of the maximum reflectivity $H_{Zm}$ generally decreases while clouds are transitioning into thunderclouds, which can be associated with the intensification of the precipitation process. This is especially evident in clouds in northwestern Russia. This process was not observed in India. Thus, thunderstorm activity occurs with the vertical development of cloud regions with large particles in Russia (according to the radar equation, one can assume the reflectivity to be strongly determined by large hydrometeors). Such an increase was not observed for post-monsoon clouds in India.

The maximum reflectivity $Z_m$ increases during the transition to the thundercloud stage for the clouds analyzed in Russia Figure 2b, which may be associated with an increase in hydrometeor sizes. In post-monsoon clouds in India, no difference in $Z_m$ distributions was noted. The maximum reflectivity was the highest in northwestern Russia in comparison with other regions at $t = -10$ min. This can be explained by the fact that the maximum reflectivity within the cloud was observed mostly below the zero-degree isotherm at this time; hence, the reflectivity was greatest in the melting layer where the ice hydrometeors are covered with a water film, which changes the dielectric properties of scattering particles and increases reflectivity.

The volumes dV35 and dV45 (not presented in Figure 2) increase with the appearance of the first discharges in the clouds under study in the Russian regions; no changes were observed in India. One of the important distinctions between post-monsoon clouds in India and continental clouds in Russia is the rarity of large ice particles in India, which significantly contribute to the reflectivity of cloud volumes above the zero-degree isotherm as noted earlier [47].

*3.2. Statistical Characteristics of Thunderclouds during the Period of LF Lightning Strikes*

The Cu characteristics derived from the radar and SEVERI instrument measurements (Meteosat satellite) are presented. These characteristics were chosen for clouds during the period of radar scans when the LF lightning discharges (further LF lightning) were observed in these clouds during the scans. In total, 380 radar scans with LF lightning strikes

were under consideration. The clouds were not differentiated according to the number of strikes, though lightning frequency was calculated for every chosen cloud and radar scan. All clouds in each of the five aforementioned groups were examined using radar, satellite, and lightning detection network data. Results of this analysis are presented in Figure 3.

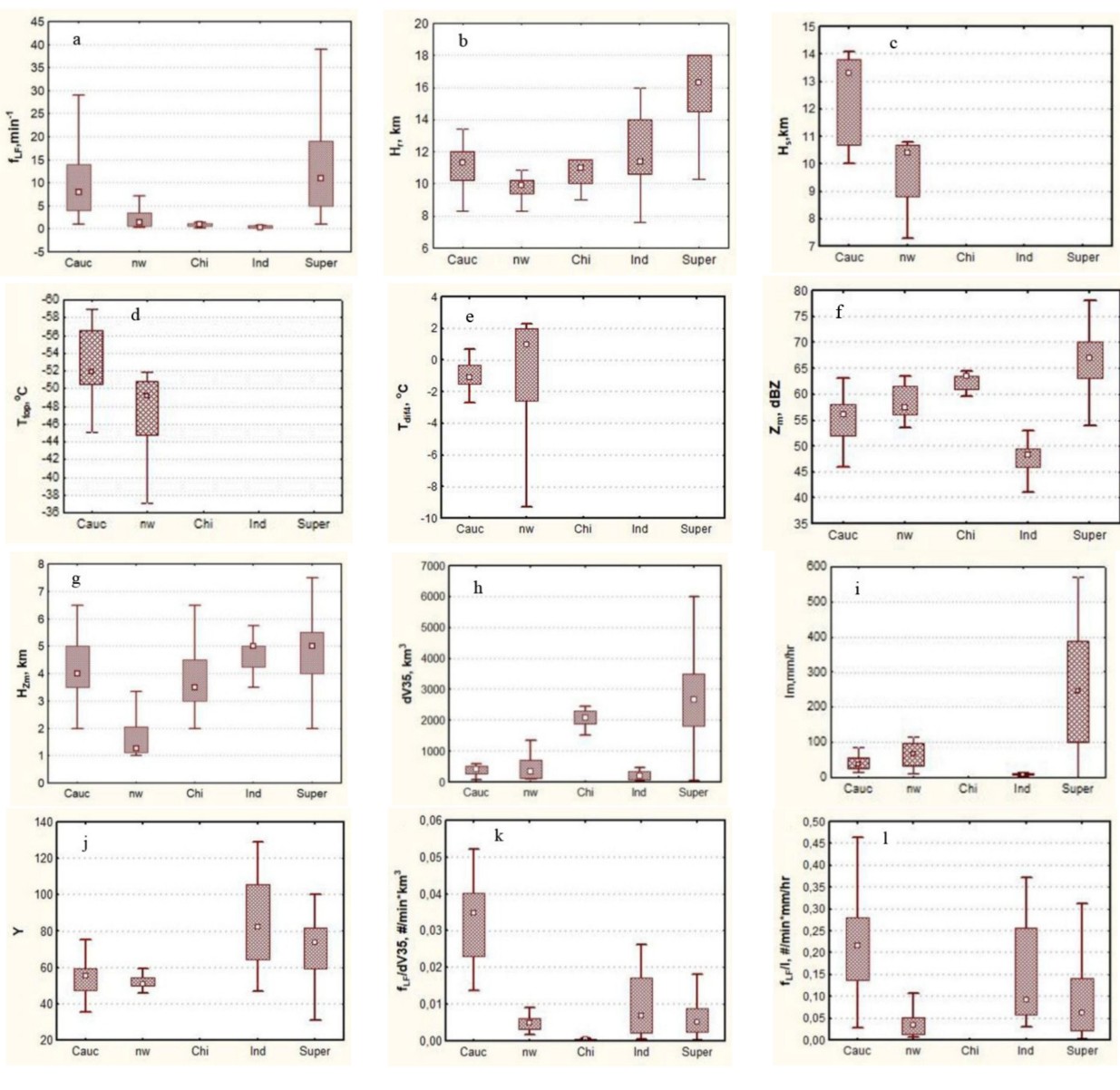

☐ Median  ▨ 25–75%  I Non-Outlier Range

**Figure 3.** Box-and-whisker diagram of the samples of cloud characteristics during the period of LF lightning strikes. Characteristics: (**a**) the frequency of lightning strikes $f_{LF}$ (min$^{-1}$); (**b**) the radar cloud top height $H_r$ (km); (**c**) the cloud top height retrieved from the SEVERI measurements $H_s$ (km); and (**d**) top temperature by SEVERI measurements $T_{top}$ °C; (**e**) difference in the brightness temperature by SEVERI measurements at 6.2 and 10.8 μm channels $T_{dif4}$ (°C); (**f**) the maximum reflectivity $Z_m$ (dBZ); (**g**) the $Z_m$ height $H_{Zm}$ (km); (**h**) the volume dV35 (km$^3$) above the zero-degree isotherm with the reflectivity higher than reflectivity threshold of 35 dBZ; (**i**) maximum precipitation intensity $I_m$ (mm/h); (**j**) thundercloud criteria $Y$; (**k**) relationship of lightning frequency to dV35 and $I_m$ ($f_{LF}$/dV35(min$^{-1}$/km$^3$; and (**l**) $f_{LF}/I_m$ (min$^{-1}$/mm/h)). The abscissa axis: north Caucasus (Cauc), northwestern Russia (nw); China (Chi); India (Ind); and supercell Cb observed in the north Caucasus (Super).

The frequency of lightning strikes $f_{LF}$, as shown in Figure 3a, greatly differs among the analyzed groups. The mean frequency was equal to 12.5 min$^{-1}$. The maximum frequency

was registered in Super-Cau_Gr, with the median being 11 min$^{-1}$, with a maximum of 99 min$^{-1}$. The lowest LF flash rates were registered in Ind_Gr, where the median was equal to 0.4 min$^{-1}$ and the maximum did not exceed 2.4 min$^{-1}$. Of course, $f_{LF}$ frequency strongly depends on the types of lightning detection systems and cloud sizes. There were no comparisons of the lightning detection systems located in the compared regions. This can contribute to some errors, though one can expect that powerful cloud-to-ground strikes are measured with reasonable accuracy by all of them. Of course, cloud sizes significantly contribute to the results.

To assess this factor, the relationship of $f_{LF}$ to dV35 was analyzed, as shown in Figure 3k. The largest median of the relationship was observed for N-Cauc_Gr and was equal to 0.034. For Ind_Gr, the relationship was 0.006; for Super-Cau_Gr, the relationship was 0.005; for N–W_Gr, the relationship was 0.004; and for Chi_Gr, the relationship was 0.0004 #/min*km$^3$. There can be some overestimation of the dV35 volume in the case of China as it was a mesoscale cloud system—and neighboring clouds and dense anvil can contribute to this volume. Thus, it can be stated that, on average, one lightning strike per minute is formed when the supercooled volume with reflectivity greater than 35 dBZ is equal to ~200 km$^3$.

Radar measurements of cloud top heights $H_r$, as shown in Figure 3b, have demonstrated that they significantly differ. On average, lightning was observed if the height of cloud top exceeded 9.5 km. Medians in the top height distributions for the groups: Ind_Gr, N-Cauc_Gr, and Chi_Gr are nearly the same and fall between 11 and 11.3 km. The biggest median was registered for Super-Cau_Gr, at 16.3 km, and the lowest for the clouds of N–W_Gr at 9.8 km. This could be expected due to the decrease in tropopause heights at higher latitudes. A minimum top height for clouds producing lightning equaled ~7 km.

The available cloud top height data from satellite measurements and rawisonde data for the N-Cauc_Gr and N–W_Gr confirm the previous conclusions and demonstrate that cloud top height is greater than 10 km when LF lightning occurs, in most cases, as shown in Figure 3c. Cloud top temperature (median value) for these groups was −51.8 and −49.2 °C, respectively, and the warmest cloud top temperature when lightning was registered was-34.5 °C (N–W_Gr), but in most cases, <−45 °C, as shown in Figure 3d.

The difference dif4 in cloud top radiation temperature, measured by Meteosat in the 6.2 and 10.8 μm channels, (dif4 = BT$_{6.2}$-BT$_{10.8}$, where BT is the brightness temperature in each channel), characterizes the intensity of cloud development. Large differences indicate the presence of intense updrafts. The dif4 for N-Cauc_Gr and N–W_Gr exceeds −3 °C in more than 50% of cases, as shown in Figure 3e. Hence, these data confirm that intense updrafts promote the transition from Cu to Cb.

Medians of the distribution of maximum reflectivity $Z_m$ differ greatly among the groups under consideration, as shown in Figure 3f. The maximum value of 67 dBZ, as expected, was in Super-Cau_Gr. $Z_m$ is greater than 52 dBZ in the majority of clouds. The minimum median was of 48.2 dBZ, for Ind_Gr. The minimum recorded reflectivity was 40.5 dBZ (an Ind_Gr case).

Heights of maximum reflectivity, $H_{Zm}$, as shown in Figure 3g, were 3.5–5 km—the only exception being N–W_GR, where the height of the maximum reflectivity was significantly less and near cloud base, indicating that it could be the result of large ice particles melting and producing a bright band.

The highest values of dV35 were found in Super-Cau_Gr, within the most powerful Cb, as shown in Figure 3h. This extreme value was 6620 km$^3$. High values of dV35 were also registered in China. The lowest dV35 values were in Ind_Gr, indicating low numbers of big ice particles.

The $Y$ parameter Figure 3j was large in all clouds, demonstrating that these Cu have transformed into thunderstorms; however, as shown earlier, the thresholds for such transitions are not universal and depend on the regional properties of Cu [48].

Heavy rain with hail was observed from Super-Cau_Gr, with radar-estimated measurements showing precipitation rates up to 570 mm/h, as shown in Figure 3i. Low intensity

precipitation was observed in Ind_Gr; the average intensity was 10 mm/h. The relationship of lightning frequency to maximum precipitation intensity might provide some insight into the role of precipitation in lightning formation. For the groups reviewed here, the median value of this relationship ranged from 0.035 to 0.21, changing six times among the groups shown in Figure 3l. Hence, maximum precipitation intensity in these cases does not allow much to be inferred about the role of precipitation in LF lightning formation.

### 3.3. Statistical Characteristics of Thunderclouds during the Period of VHF Lightning Strikes

The characteristics of the clouds when VHF strikes occur are presented in Figure 4. In total, 340 radar scans with VHF strikes were under consideration. The mean frequency was equal to 135.6 $\min^{-1}$. A median value of the distributions of lightning frequency $f_{VHF}$ varied very significantly, as shown in Figure 4a. It exceeded 100 $\min^{-1}$ for the clouds observed at the N-Cauc_Gr and Super-Cau_Gr, which equals 27.6 $\min^{-1}$ for the Chi_Grn, but it is less than 1 $\min^{-1}$ for the clouds in Ind_Gr. A maximum $f_{VHF}$, observed during radar scans, was 725 $\min^{-1}$ within Super-Cau_Gr. This should be expected, as this was the biggest storm of all those considered. The "normalized" frequency $f_{VHF}/\text{dV35}$ also differed significantly between the groups, as shown in Figure 4h. The frequency differed by nearly two orders of magnitude, indicating that some cloud volumes with high reflectivities did not result in significant cloud electrification.

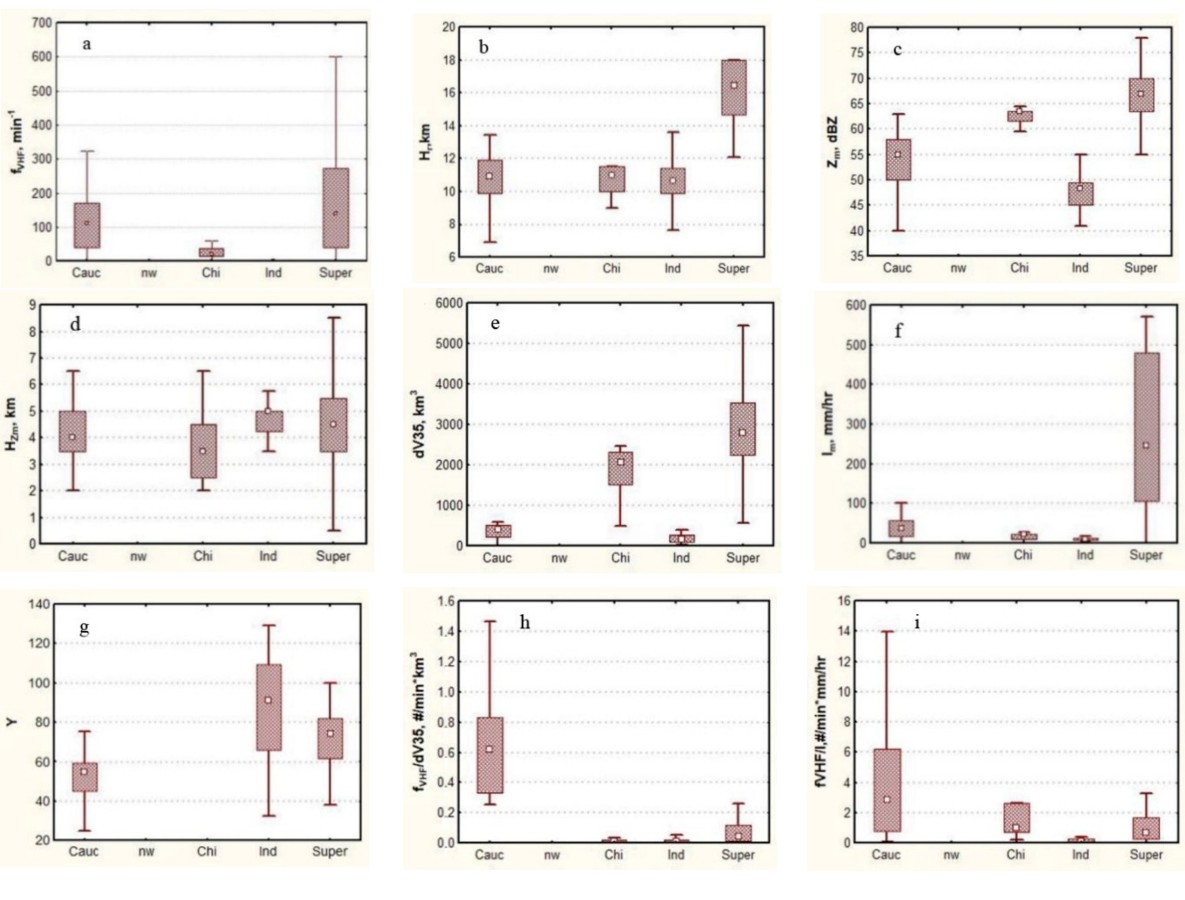

Median ▦ 25–75% ⊥ Non-Outlier Range

**Figure 4.** Box-and-whisker diagrams for storms observed when lightning was detected by VHF networks. Characteristics: (**a**) the frequency of lightning strikes $f_{VHF}$ ($\min^{-1}$); (**b**) the cloud top height $H_r$ (km); (**c**) the maximum reflectivity $Z_m$ (dBZ); (**d**) the $Z_m$ height $H_{Zm}$ (km); (**e**) the volume dV35 (km$^3$) above the zero-degree isotherm with the reflectivity higher than the reflectivity threshold of 35 dBZ; (**f**) maximum precipitation intensity $I_m$ (mm/h); (**g**) thundercloud criteria $Y$; (**h**) relationship of lightning frequency to dV35 and $I_m$ ($f_{LF}/\text{dV35}(\min^{-1}/\text{km}^3$; and (**i**) $f_{LF}/I_m$ ($\min^{-1}/\text{mm/h}$)). The abscissa axis: north Caucasus (Cauc), northwestern Russia (nw); China (Chi); India (Ind); and supercell Cb observed in the north Caucasus (Super).

The heights of the cloud tops $H_r$ are similar to those of the investigated regions shown in Figure 4b. The means and medians of $H_r$ distributions are within the range of 10.5–11 km—the only exception being supercell Cb, where the top height exceeds 16 km.

Medians of $Z_m$ distributions differ for the investigated regions and range from 48.2 to 67 dBZ, as shown in Figure 4c. Maximum reflectivity was measured in Super-Cau_Gr, and the minimum in Ind_Gr, where discharges were recorded with reflectivity as low as 33.5 dBZ. The height of maximum reflectivity ranged from 3.5 to 5 km (means and medians). The maximum median of 5 km was found in the clouds in Ind_Gr, located near the height of the 0 °C isotherm in most cases. Crystal melting took place at and below this height; thus, this was a "bright band".

Supercooled volumes with reflectivities higher than 35 dBZ vary within the range of 400–2795 km$^3$. The supercell Cb had the biggest supercooled volume, likely because of large ice hydrometeors in the upper portion of the cloud. The lowest values were found in the Ind_Gr clouds, as shown in Figure 4e.

Measurements of rain intensity show that the most intense precipitation was observed in the Super-Cau_Gr, while only relatively light rain was observed in the Ind_Gr clouds, as shown in Figure 4f. Note that radar measurements of precipitation intensity can have significant errors when large hail is present [45], as was observed in the case of Super-Cau_Gr.

The $Y$ parameter was high in all groups (N-Cauc_Gr, Super-Cau_Gr, Ind_Gr), indicating a high probability of lightning flashes in most cases, as shown in Figure 4g.

The relationship of $f_{VHF}/I$ (median) ranged from 0.1 to 2.8 #/min*mm/h, as shown in Figure 4i. The variability reconfirms the previous statement that the maximum precipitation intensity does not leave much to be inferred about the role of precipitation in lightning formation.

### 3.4. Relations of Frequency of LF Discharges with Supercooled Volume and Precipitation Intensity

The Spearman correlation coefficient r between $f_{LF}$ (frequency of cloud-to-ground lightning strikes) and dV35 varies widely, from 0.06 to 0.90.

For this analysis, all clouds can be categorized as belonging to one of three types. The first type is the single-cell or isolated Cb. The following cases are of this type: Pyatigorsk, Gulf of Finland, 22 July, Ladoga Lake, Petrokrepost, and China. The correlation coefficient, $r$, for this type of clouds, varies between 0.21 and 0.90.

The second type is the mesoscale convective system (Kislovodsk case) and Supercell case. The correlation coefficient is less than 0.23 for these clouds. It was anticipated that $r$ would be less than for the first type of clouds, as these cells exist as cumulonimbus for different fractions of their lifetimes.

The third type consists of multiple Cu. These include two cases in Maharashtra, India, ($r = 0.39$) and Karnataka, India ($r = 0.51$).

The correlation coefficients for linear relations between $f_{LF}$ and dV35 are presented in Table 2 for the cases with $r > 0.5$ ($f_{LF} = A + B * dV35$, $f_{LF}$ in min$^{-1}$, dV35 in km$^3$).

**Table 2.** Correlation coefficients and constants for the equations relating low frequency flashes to changes in the 35 dBZ cloud volumes, for cloud groupings. In each case, $f_{LF} = A + B * dV35$. The abbreviations shown for each cloud group are those used in Figure 5.

| Case | Pyatigorsk (Pat) [31] | 22 July (SPb) [33,36] | Gulf of Finland (Fin) [31,35] | Karnataka (Karn) | China [1] |
|---|---|---|---|---|---|
| $r$ | 0.66 | 0.71 | 0.90 | 0.51 | 0.67 |
| A | −0.012 | −0.091 | −1.04 | 0.1 | −0.56 |
| B | 0.0085 | 0.0046 | 0.0086 | 0.0031 | 0.0007 |

China significantly differed from other cases, showing that very large volumes are required to form lightning. The reason was discussed above; hence, it was excluded from the analysis below. The relationships are also illustrated in Figure 5. The difference between them is relatively small, especially considering that data were obtained for different regions

with radars of different types and especially the differences in the lightning detection systems. The greatest lightning frequencies were observed in the case of Pyatigorsk (north Caucasus, Russia) and the lowest for the case of Karnataka (India).

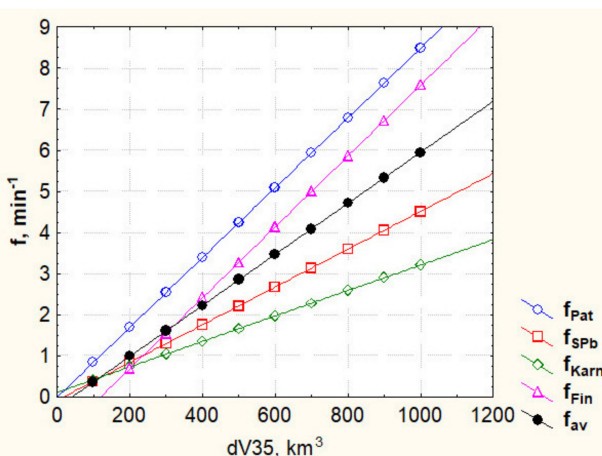

**Figure 5.** Linear approximations between the LF strikes frequency $f$ (min$^{-1}$) and supercooled volume dV35 ($f_{av}$—averaged frequency). Abbreviations are as indicated in Table 2; $f_{av}$ is the mean relationship for all four cloud groups.

The relationships were averaged, resulting in a general equation for all clouds except those observed in China:

$$fav = -0.26 + 0.0062 * dV35 \tag{7}$$

where $f$ is the lightning frequency (cloud-to-ground) in min$^{-1}$ and dV35 (km$^3$) is the supercooled volume of the cloud having a radar reflectivity more than 35 dBZ and above the height of the 0 °C isotherm (km$^{-3}$).

The data show that for any given cloud volume, the lightning frequency can vary by up to a factor of 2.6, as shown in Figure 5. Hence, the average equation should only be used for the assessment of lightning frequency and should be verified for each region, lightning detection system, and radar.

The intensity of precipitation reflects the microphysical processes in the cloud. High intensities, such as those often observed in Cb, result from the fallout of big ice hydrometeors, including hail and their melting. The presence of large ice particles enables interactions with cloud crystals that result in Cb electrification [6]. Therefore, there should be a correlation between precipitation intensity and cloud electrification, and as a result, the frequency of lightning flashes [49].

The correlation coefficients for the cases under consideration range from 0.09 to 0.55. Lightning frequency depends less on the maximum precipitation intensity than dV35. There is only one case with $r > 0.5$, which is that of Karnataka. We conclude that in most cases, lightning frequency is not correlated with maximum precipitation intensity and should not be used for the estimation of flash frequencies.

*3.5. Relations of Frequency of VHF Discharges with Supercooled Volume and Precipitation Intensity*

Some modern lightning detection systems provide the capability to detect intra-cloud and cloud-to-cloud discharges, which are prevalent during thunderstorm lifecycles. VHF strikes mostly characterize intra-cloud and cloud-to-cloud discharges [41]. The LS-8000 network in Russia and lightning detection system of the Indian Institute of Tropical Meteorology (IITM) in India were used to study the discharges in N-Cauc_Gr and in Ind_Gr, respectively.

The correlation coefficients between $f$ and dV35 were calculated. There were four cases, where $r > 0.5$: Pyatigorsk ($r = 0.71$), Karnataka ($r = 0.6$), Maharashtra ($r = 0.74$), and China ($r = 0.83$). Though correlations were significant, the relationships between $f$ and

dV35 differed greatly. The difference in the number of flashes per similar dV35 in the cases of Pyatigorsk, India, and China reached orders of 1–2. Hence, we conclude that there are no predictable relationships between these parameters.

The analysis shows that correlation coefficients larger than 0.5 between *f* and *I* were only found in one case: Karnataka ($r$ = 0.51). As correlation coefficients were less than 0.5 in other cases, we conclude that there are no consistent relationships.

### 3.6. Lightning Currents and Transferred Charges

Lightning currents were measured and analyzed for the following groups: Ind_Gr, Cauc_Gr, and Super-Cau_Gr. The data available for these two regions differed. Every strike current (in LF or VHF range) was analyzed when lightning was recorded in Ind_Gr. No positive discharges were measured in Ind_Gr in the LF range. Maximum currents (positive I+ and negative I−) were available for the duration of each radar scan in the N-Cauc_Gr and Super-Cau_Gr, but only for LF strikes. Box-and-whisker plots of the resultant distributions are presented in Figure 6.

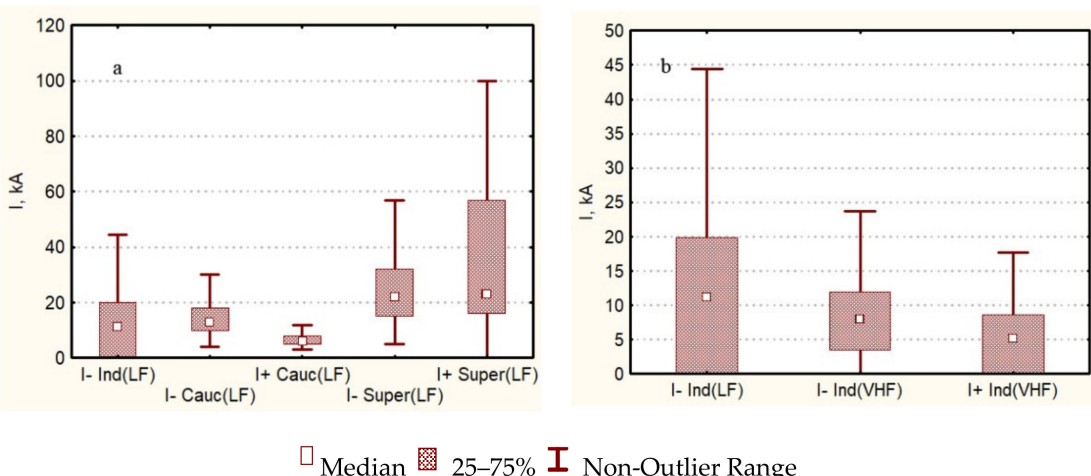

**Figure 6.** Box-and-whisker diagram of samples of flash current *I* (kA) in the LF range (left diagram (**a**)) and in the LF and VHF range (right diagram (**b**)). The abscissa axis (groups of clouds): India (I− Ind (LF)); north Caucasus (I− Cauc (LF)); north Caucasus (I+ Cauc (LF)); supercell Cb (I− Super (LF)); supercell Cb (I+ Super (LF)); India (I− Ind (VHF)); (I− Ind (VHF)). I− and I+ indicate the maximum currents (positive I+ and negative I−).

For the LF range, the mean and median values of the maximum (by modulus) negative currents do not differ much between lightning in the Ind_Gr and N-Cauc_Gr, as shown in Figure 6a. The observed currents in Super-Cau_Gr were significantly greater, however, with a median of −22 kA. The maximum (by modulus) current was 67 kA. Positive currents were greater than negative in the Super-Cau_Gr, and vice versa in N-Cauc_Gr. The maximum positive current was measured in Super-Cau_Gr and equaled 140 kA. The observed currents in the VHF range shown Figure 6b in Ind_Gr (median values −8.0 and +5.1 kA) were less than those registered in the LF range (median −11.1 kA).

Transferred charges were measured in the LF range for lightning in Pyatigorsk thunderstorms. Lightning can transfer positive or negative charge from the cloud to the Earth. A sum of negative lightning currents for a period of cloud existence was equal to −3874 kA, and positive to +726 kA. The maximum current of positive lightning was equal to 110 kA, and negative −22 kA. Currents are usually calculated using the magnitude of the electrical field pulse during the return stroke (main discharge) of lightning [50]. This stage was characterized by a short time and maximum current. To calculate the transferred charge, one integrates the current throughout the discharge. The data for such calculations were unavailable, hence the simplified procedure was used. Each charge was calculated by multiplying the current recorded during the return stroke by 0.1 s (one-tenth of a second).

This is 5 to 10 times less than the average duration of a typical lightning flash. With this approximation, the total charge transferred from the cloud by negative lightning was $-387.4$ C, with an average charge per flash of $-0.44$ C. The corresponding charges transferred by positive lightning were 72.6 and 0.60 C, respectively. Thus, lightning from the chosen thunderstorm delivered a total negative charge of approximately $-300$ C.

## 4. Discussion

The clouds reviewed developed in India, China, and two regions of Russia—north Caucasus and northwestern Russia in the vicinity of St. Petersburg. It was believed that different physical, geographical, and climatic conditions would influence the development of thunderclouds. Several field projects were discussed. Five groups of clouds were analyzed, which were separated according to geographical position and cloud characteristics. Four groups corresponding to the geographical regions, and one special "group", a supercell in the north Caucasus, were expected to have extreme characteristics.

The atmosphere parameters were typical for the development of continental type clouds in Russia and China. India stands out from this series because of the monsoon circulation, which is typical for this region.

Cloud characteristics were obtained with the help of measurements by radars, the SEVERI instrument on the Meteosat satellite, and several lightning detection systems. Radar reflectivity depends on the presence of big particles to a great extent. Cu electrification also depends on the interaction of graupel with ice crystals; hence, it could be expected that intensive electrification and lightning flashes will be found in the cases with high reflectivity. This was based upon numerous observations, and referred to in the introduction. Regional peculiarities of lightning formation remain a subject of interest [51,52].

Cu characteristics during their initial stage of transition to thunderclouds were studied. Cloud characteristics were compared during two consecutive radar scans: just before any flashes were observed, and during the first occurrence of flashes (within 10 min).

In the Russian regions under study, the altitudes of the maximum reflectivity $H_{Zm}$ generally decrease while clouds are transitioning to thunderstorms, which can be associated with the intensification of the precipitation process. This process was not observed in India. The maximum reflectivity, $Z_m$, increases during the transition to the thundercloud stage for the clouds analyzed in Russia, which may be associated with an increase in hydrometeor sizes. In post-monsoon clouds in India, no difference in $Z_m$ distributions was noted. Lightning strikes were accompanied by a noticeable increase in dV35 in the Russian regions, while no changes were observed in India. One of the important distinctions between post-monsoon clouds in India and continental clouds in Russia was the rarity of large ice particles (in India), which to a large extent determined the reflectivity of a cloud above the zero-degree isotherm, as noted previously [47].

The cloud top height was the only characteristic of clouds which increased during their transition to the thundercloud stage in all regions. It is the highest (the median of distribution is 13.5 km) in India and the smallest (the median is 9.5 km) in northwestern Russia, corresponding to a general increase in the depth of clouds (height of the tropopause) with a decrease in latitude.

Statistical characteristics of the clouds were studied at the period of radar scans, when LF lightning was observed just during the scans. The frequency of lightning strikes $f_{LF}$ differed greatly among the analyzed groups. The maximum frequency was registered in Super-Cau_Gr; the median was equal to 11 min$^{-1}$, and the maximum was 99 min$^{-1}$, which fits the published data on the frequency of flashes in such clouds, which can amount to hundreds of flashes per minute [4]. High correlation between lightning frequency $f_{LF}$ and dV35, observed in most cases, made it possible to assess the supercooled cloud volume at the time lightning flashes begin; it was equal to ~200 km$^3$. High correlations between the radar reflectivity at altitudes above the 0 °C isotherm and the lightning flash rate may be informed by the non-inductive charging theory. This holds that collisions among radar-sensitive, precipitating ice particles, and smaller, cloud ice particles may separate

the electric charge when supercooled liquid water is present. A strong correlation between flash frequency and the supercooled volume and cloud area for the temperatures less than $-5\,°C$ was also obtained in previous experiments [18,52].

Radar measurements of cloud top heights $H_r$ demonstrated that they differ. On average, lightning flashes were observed if the cloud top height exceeded 9.5 km. Medians in top height distributions for the groups: Ind_Gr, N-Cauc_Gr, Chi_Gr were within 11–11.3 km. The biggest median value was registered for Super-Cau_Gr, that being 16.3 km, and the lowest, for the clouds of N–W_Gr, at 9.8 km. This could be expected due to decrease in tropopause heights at higher latitudes. The minimum top height for clouds producing lightning was ~7 km. In agreement with previous studies [18,51], these data show that the echo top height needed for Cu to transition to Cb differed depending on the region and cloud character, and can be used for simple flash rate assessments. The maximum cloud top temperature when lightning was recorded was equal to $-34.5\,°C$ for N–W_Gr, but in most cases, was $<-45\,°C$.

Medians of the distribution of maximum cloud reflectivity $Z_m$ differed greatly. The maximum value of 67 dBZ, as expected, was in Super-Cau_Gr. The minimum median of 48.2 dBZ was observed for Ind_Gr. $Z_m$ is greater than 52 dBZ in most of the clouds studied. An exception is that of the clouds in Ind_Gr, where lightning was observed when maximum reflectivity in most clouds exceeded 45 dBZ. The minimum recorded reflectivity was 40.5 dBZ (Ind_Gr case). The difference in reflectivity for clouds over land and ocean was mentioned earlier, also in the investigations [7]. The relationship of lightning frequency to maximum precipitation intensity can characterize the role of precipitation in formation of lightning. For the cloud groupings under study here, this relationship changed six times among the groups.

The statistical characteristics of the clouds at the moment of the occurrence of VHF flashes were considered. The median value of the distributions of lightning frequency $f_{VHF}$ varied very significantly. It exceeded $100\,min^{-1}$ for the clouds observed at the N-Cauc_Gr and Super-Cau_Gr, was $27.6\,min^{-1}$ at the Chi_Grn, but was less than $1\,min^{-1}$ for the clouds in Ind_Gr. A maximum $f_{VHF}$, observed during radar scans, was $725\,min^{-1}$ at Super-Cau_Gr. The "normalized" frequency $f_{VHF}/dV35$ also significantly differed among the groups; some cloud volumes with high reflectivity were not the source of significant cloud electrification. Wide variations in the $f_{VHF}/I$ and $f_{LF}/I$ relationships show that these do not provide as much insight regarding the role of precipitation in lightning formation. Precipitation flux may be more reasonable for this task.

Relations of the frequency of LF discharges with supercooled volume and precipitation intensity were analyzed. The averaged equation between $f_{LF}$ and dV35 was derived and can be used for $f_{LF}$ assessments. No consistent relationships between $f_{VHF}$ and dV35 or $I$ were discovered.

Lightning currents and transferred charges were measured and assessed. Maximum currents were observed in Super-Cau_Gr. A maximum negative current was $-67$ kA, and positive current was $+140$ kA. The currents are rather big and exceed those recorded in China by 2–4 times [53]. The observed currents in the VHF range were less than those in the LF range. This is because the current in VHF flashes characterize mostly intra-cloud discharges. Transferred charges were assessed in the LF range for the lightning observed in Pyatigorsk. The charges transferred from the cloud by negative lightning totaled 387.4 C, with an average charge per flash of 0.44 C. Similarly, the charge transferred by positive lightning was 72.6 and 0.60 C, respectively.

## 5. Conclusions

A comparison of thundercloud characteristics in different regions of the world was conducted. The clouds studied developed in India, China, and in two regions of Russia.

Cu characteristics during their initial stage of transition to thunderclouds were analyzed. The cloud top height was the only characteristic of the clouds which increased during their transition to the thundercloud stage in all the regions. The maximum reflectiv-

ity, $Z_m$, and the volume, dV35, increased during the transition to the thundercloud stage for the clouds analyzed in Russia. In the post-monsoon clouds in India, no difference in $Z_m$ and dV35 distributions was noted.

The statistical characteristics of the clouds were also studied during radar scans, when LF lightning discharges were observed. The frequency of lightning strikes $f_{LF}$ differed greatly among the analyzed groups. The maximum frequency was registered in a supercell Cb, which developed in the Caucasus; the median of distribution was −11 min$^{-1}$ and the maximum was 99 min$^{-1}$. Measurements of cloud top heights showed that the echo top height needed for Cu to transition to Cb differ, depending on the region and cloud character. On average, lightning was observed if the height of cloud top exceeded 9.5 km. Medians in top height distributions for clouds in India, North Caucasus, and China were nearly the same and fall between 11 and 11.3 km. The biggest median was registered for Super-Cau_Gr, 16.3 km, and the lowest for the clouds of N–W_Gr at 9.8 km. On average, lightning flashes were observed if the cloud top height exceeded 9.5 km.

The medians of the distribution of maximum cloud reflectivity $Z_m$ differed greatly. The minimum median of maximum cloud reflectivity $Z_m$ of 48.2 dBZ was observed for clouds in India. $Z_m$ was greater than 52 dBZ in most of the other clouds studied. Post-monsoon clouds in India differ from those of other regions. Lightning strikes are accompanied by a noticeable increase in dV35 in the Russian regions, while no changes were observed in India. One of the important distinctions between post-monsoon clouds in India and continental clouds in Russia is the rarity of large ice particles (in India), which determine to a large extent the reflectivity of a cloud above the zero-degree isotherm.

Statistical characteristics of the clouds when VHF flashes occurred were considered. The median value of the distributions of lightning frequency $f_{VHF}$ varied very significantly. It exceeded 100 min$^{-1}$ for the clouds observed at the north Caucasus, is 27.6 min$^{-1}$ in the Cb in China, but it is less than 1 min$^{-1}$ for the clouds in India.

The relationships between the frequency of LF discharges and supercooled volume were analyzed. An equation describing the relationship between $f_{LF}$ and dV35 was derived and can be used for $f_{LF}$ assessments. On average, one lightning strike is formed per minute when a supercooled volume with a reflectivity greater than 35 dBZ is equal to ~200 km$^3$.

Lightning currents were measured. Maximum currents were observed in the one supercell Cb. The maximum negative current was −67 kA, and the maximum positive current was +140 kA.

**Author Contributions:** Conceptualization, A.S.; methodology, B.B., S.P., J.Y.; formal analysis, A.A., Y.M., J.G., M.T.; investigation, V.G., Y.D., A.K., N.V.; resources, A.S., B.B., S.P., J.Y.; writing—original draft preparation, A.S. All authors have read and agreed to the published version of the manuscript.

**Funding:** The research was supported by the Russian Foundation for Basic Research (grant BRIKS_t 18-55-80020), China grant 2018YFE101200 and Department of Science and Technology, Government of India (BRICS/PilotCall2/LiESTER/2018(G)).

**Data Availability Statement:** Detailed information about projects where data were obtained and summaries of the Cu characteristics observed in each case were presented in our earlier publications [1,29–39].

**Acknowledgments:** Data were collected thanks to the support of the Main Geophysical Observatory and High-Mountain Geophysical Institute, Russia; Weather Modification International, USA; Indian Institute of Tropical Meteorology; and the Institute of Atmospheric Physics, China.

**Conflicts of Interest:** The authors declare no conflict of interest.

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
