# Peer review of "Investigation of Thundercloud Features in Different Regions"

_remotesensing, doi:10.3390/rs13163216_

Round 1

Reviewer 1 Report

This work gives a comparison of thundercloud characteristics in different regions of the world. As an empirical study, this paper gives some interesting conclusions. I think it's a good article that deserves attention. However, some aspects of the paper still need to be re elaborated or modified.
1. In the introduction, not only the relationship between lightning frequency and updraft is introduced, but also the lightning detection system is introduced, which seems not related to the characteristics of thunderstorms in different areas studied in this paper. It is suggested to describe the significance and value of the survey and its outstanding contribution to this field.
2. Are the “section 2", “section 3", “section 4", “section 5", “section 6", “section 7", “section 8", “section 9" and “section 10.2" mentioned in lines 76 to 84 wrong? There are only five section in the manuscript.
3. Line 198 "data in India were obtained during the period from late segment to early October 2019." and line 201 "in northern Russia were obtained for the periods from may – segment in 2017 and 2018." the amount of data collected is small, only one or two years of data is difficult to get the general rule. In addition, for the 477 line discussion part, the description data comes from the four parts "India, China and two regions of Russia: Northern caucus and northwest Russia", but the actual data collection does not have China, so the full text description should be consistent.
4.What does section 9, section10.2 refer to ? line76-84
5.The main contribution of this article is unclear, so it is recommended to summarize the innovation points at the end of the introduction.
6.Please arrange the citations in order. The citations are confused.
7.Figure 1. is not clear enough and the font is not clear.
8.Each box-and-whisker in Figure 2.3.4.6. should corresponds to subtitles.

Author Response

Reviewer #1:

This work gives a comparison of thundercloud characteristics in different regions of the world. As an empirical study, this paper gives some interesting conclusions. I think it's a good article that deserves attention. However, some aspects of the paper still need to be re elaborated or modified.
Response: We thank the reviewer very much for his/her positive review. These insightful comments and suggestions are very helpful to improve our manuscript. Our point-by-point responses are listed below.

  1. In the introduction, not only the relationship between lightning frequency and updraft is introduced, but also the lightning detection system is introduced, which seems not related to the characteristics of thunderstorms in different areas studied in this paper. It is suggested to describe the significance and value of the survey and its outstanding contribution to this field.

Response: We have added a paragraph at the end of introduction –

The presented  brief review identifies Cb characteristics typical to thundercloud-transition stage of Cu development,  important for understanding of cloud electrification physics. This also  has practical implications for lightning prediction.

Attempts to establish relationships between lightning and the dynamic and microphysical cloud characteristics have been primarily made through remote sensing in recent years. Lightning characteristics, including lightning frequency, depend on Cb properties. It follows from the literature review that crystal phase likely plays a role in lightning formation. Those specific habits of crystal phase most important for Cu electrification remain a challenge, and remain a topic  for future investigations. At the same time, the varied conditions in which, thunderclouds develop, suggest that Cu electrification and hence, lightning formation, might result from differing mechanisms. This statement is supported by the regional variation in Cb properties when  lightning is first observed.

The objectives of the present paper are to retrieve thundercloud characteristics and to quantify the interrelations between the parameters of lightning strikes and Cu characteristics during lightning events in different parts of the world. It is postulated that thundercloud properties are differ significantly in the varying climate  under study: north and south  Russia, India and China. Hence, one can expect that relations between lightning characteristics and Cu characteristics will also be different, suggesting that differences in Cu electrification play an important role in lightning initiation frequency. Special attention is paid to lightning formation in a supercell Cb, that developed in the Northern Caucasus of Russia.

  1. Are the “section 2", “section 3", “section 4", “section 5", “section 6", “section 7", “section 8", “section 9" and “section 10.2" mentioned in lines 76 to 84 wrong? There are only five section in the manuscript.

Response: Yes, they are mentioned in wrong way, sorry for inconvenience, these sections were in draft version. Changes in the manuscript have been made.

  1. Line 198 "data in India were obtained during the period from late segment to early October 2019." and line 201 "in northern Russia were obtained for the periods from may – segment in 2017 and 2018." the amount of data collected is small, only one or two years of data is difficult to get the general rule. In addition, for the 477 line discussion part, the description data comes from the four parts "India, China and two regions of Russia: Northern caucus and northwest Russia", but the actual data collection does not have China, so the full text description should be consistent.

Response: We agree with you, the presented data does not form large amount of statistical units, hence we state in the article-

There is need to mention that comparison of the Cu characteristics is based on available data and includes different numbers of clouds and observations, hence it does not pretend to be a true climatological study, but the data processing is done in the same manner, and the results show trends in Cu characteristics and their differences, depending on the geographical region.

We would like to mention, that for most of the analyzed cloud characteristics were obtained during a period from their origin up to their decay. Very thorough analysis of every case was carried out, every lightning stroke has been matched with Cu position.  Results were published in several articles. This approach is different from regularly used statistical procedures aimed to get data. From our point of view, we have highly reliable data.

We have rather limited number of measurements in China, so we compare  Cb –lightning properties  with Cb in other  regions rather carefully and only for limited number of characteristics.

4.What does section 9, section10.2 refer to ? line76-84

Response: They are mentioned in wrong way. Changes in the manuscript have been made.

5.The main contribution of this article is unclear, so it is recommended to summarize the innovation points at the end of the introduction.

Response: We have added a paragraph at the end of introduction –

The following data illustrate a significant difference between thundercloud characteristics in Russia and China, in comparison with India. Moreover, thundercloud characteristics also differ for northern and southern Russian regions. These features can be explained by differing troposphere depth, the height of zero-degree isotherm, and other troposphere properties: temperature, humidity, aerosol content. The observations confirm the importance of crystal phase in Cu electrification. A volume with strong radar reflectivity above zero-degree isotherm, a prerequisite for lightning formation, is assessed.

6.Please arrange the citations in order. The citations are confused.

Response: In accordance with the accepted rules of Remote Sensing the citations should be in order of their references in the text, we follow the rules.

7.Figure 1. is not clear enough and the font is not clear.

Response: We agree that this strict climatologic statements and hence Fig.1,  do not suite main ideas of the article. We have changed the figure to the new one, which demonstrates the position of the regions, where measurements were carried out, the climatological part is shortened and is presented as a text. 

8.Each box-and-whisker in Figure 2.3.4.6. should corresponds to subtitles.

Response: Changes in subtitles have been made.

 Thanks for your time,

Authors

Reviewer 2 Report

The manuscript aims to investigate thunderstorms observed with radar and lightning flashes observed in their vicinity. Unfortunately neither the methods nor the data are adequately described. Therefore it is not possible for me to make an assessment of the quality of the scientific findings. I read the first six pages of the manuscript but since crucial information is missing I don't see much value in continuing further as I don't have the information I would need to make informed judgments. I therefore recommend to reject the manuscript in its current form. I encourage the authors to write a substantial Data and Methods section they can refer to throughout the paper. Only once this is accomplished will reviewers be able to evaluate the rest of the manuscript.

The specific comments for the first six pages are below.

1. Introduction: The authors nicely describe the previous literature on the topic. However, they do not make the connection from the literature to their research objective. I.e. they do not clearly identify the gaps in the literature they are trying to fill.

2. Line 32: Who is "most"? Consider re-wording.

3. Line 92: The definition of Dfb given here is different from the one at line 94.

4. Figure 1 is impossible to read, especially the text and is therefore rather useless in its current form. In addition, I don't believe that the climatic region is the most important information to show on these maps as only individual clouds and not a climatology is studied. I suggest the following: a) Add one more panel with an overview map which shows the general locations of the regions. b) Replace the climate classification with topography which would be more interesting from a meteorological point of view. c) Show the locations of the clouds described later and the locations of the radars on the maps.

5. Section 2.2: There is almost no information on the radar data set and the datasets are not cited. Where is the data available? What is the scan strategy of the radars? How frequently are scan volumes provided? What is the elevation and azimuth angle spacing? What is the spacial and temporal resolution of the data? What is meant with "Beam angle" in Table 1? Which variables are used? Etc.

6. There is even less information on the lightning data. It needs to be described in detail.

7. Line 159: What are "statistically significant volumes of data"?

8. What is Y called and what does the cr stand for in Ycr?

9. Lines 132-165: What data is used to identify the isotherms? It needs to be described in detail as well.

10. Line 166: It is unclear what is meant with thunderstorm occurrence. Do you mean that an existing storm will develop lightning activity, turning from a "regular" storm into a thunderstorm?

11. Please provide references for equations 2-6. Please also clarify if Z, Zw, and Zi are in linear or logarithmic units and how they are different from the Z3 mentioned before.

12. What are b and bw in equation 3? What are the bins mentioned in line 176?

13. References for the "previous investigations" are missing at line 192.

14. Starting line 197: The description of the clouds should be in the "Methods" section.

15. Line 204: What were the criteria for selecting the 60 clouds?

16. In the description of the studied cloud groups (line 109) only very few clouds are mentioned but now 60 clouds in each region are described. This is very confusing.

17. Line 205: Since we know nothing about the timing and time resolution of the lightning and the radar data it is unclear how the time "10 minutes before lightning onset" is determined. All of this information belongs in the "Methods" not the "Results" section.

18. Line 207: 5 dBZ does not represent cloud top height.

19. Line 208: How is the maximum reflectivity defined? Of the whole cloud? In a vertical column?

20. Line 211: Since we don't even know if the radar data is in polar or Cartesian coordinates, or what their resolution is, how are we supposed to interpret "Radar data ... were horizontally and vertically averaged in 1 km steps."?

21. Line 231: How is the "lightning frequency distribution" defined?

Etc.

Author Response

Reviewer #2:

1.The manuscript aims to investigate thunderstorms observed with radar and lightning flashes observed in their vicinity. Unfortunately neither the methods nor the data are adequately described. Therefore it is not possible for me to make an assessment of the quality of the scientific findings. I read the first six pages of the manuscript but since crucial information is missing I don't see much value in continuing further as I don't have the information I would need to make informed judgments. I therefore recommend to reject the manuscript in its current form.

Response: We thank the reviewer very much for his/her review. The following insightful comments and suggestions are very helpful to improve our manuscript. We are sorry for not too clear presentation and tried to improve these shortcomings in the revised version.

Our point-by-point responses are listed below.

  1. I encourage the authors to write a substantial Data and Methods section they can refer to throughout the paper. Only once this is accomplished will reviewers be able to evaluate the rest of the manuscript.

Response:Data and Methods section was rewritten, the needed data was added.

  1. Introduction: The authors nicely describe the previous literature on the topic. However, they do not make the connection from the literature to their research objective. I.e. they do not clearly identify the gaps in the literature they are trying to fill.

Response: We added the necessary information at the end of Introduction-

The presented  brief review identifies Cb characteristics typical to thundercloud-transition stage of Cu development,  important for understanding of cloud electrification physics. This also  has practical implications for lightning prediction.

Attempts to establish relationships between lightning and the dynamic and microphysical cloud characteristics have been primarily made through remote sensing in recent years. Lightning characteristics, including lightning frequency, depend on Cb properties. It follows from the literature review that crystal phase likely plays a role in lightning formation. Those specific habits of crystal phase most important for Cu electrification remain a challenge, and remain a topic  for future investigations. At the same time, the varied conditions in which, thunderclouds develop, suggest that Cu electrification and hence, lightning formation, might result from differing mechanisms. This statement is supported by the regional variation in Cb properties when  lightning is first observed.

The objectives of the present paper are to retrieve thundercloud characteristics and to quantify the interrelations between the parameters of lightning strikes and Cu characteristics during lightning events in different parts of the world. It is postulated that thundercloud properties are differ significantly  in the varying  climate  under study: north and south  Russia, India and China. Hence, one can expect that relations between lightning characteristics and Cu characteristics will also be different, suggesting that differences in Cu electrification play an important role in lightning initiation frequency. Special attention is paid to lightning formation in a supercell Cb, that developed in the Northern Caucasus of Russia.

The following data illustrate a significant difference between thundercloud characteristics in Russia and China, in comparison with India. Moreover, thundercloud characteristics also differ for northern and southern Russian regions. These features can be explained by differing troposphere depth, the height of zero-degree isotherm, and other troposphere properties: temperature, humidity, aerosol content. The observations confirm the importance of crystal phase in Cu electrification. A volume with strong radar reflectivity above zero-degree isotherm, a prerequisite for lightning formation, is assessed.

  1. Line 32: Who is "most"? Consider re-wording.

Response:Most is changed on most scientists

  1. Line 92: The definition of Dfb given here is different from the one at line 94.

Response:Changers were made. This climatological part of the paper was excluded and the new text was written.

  1. Figure 1 is impossible to read, especially the text and is therefore rather useless in its current form. In addition, I don't believe that the climatic region is the most important information to show on these maps as only individual clouds and not a climatology is studied. I suggest the following: a) Add one more panel with an overview map which shows the general locations of the regions. b) Replace the climate classification with topography which would be more interesting from a meteorological point of view. c) Show the locations of the clouds described later and the locations of the radars on the maps.

Response: We agree, these plots do not contribute much to the investigation and are not too clear for understanding. Your idea to present location of areas, where measurements have been carried out, is valuable. So, Fig 1 was changed by the map, a new text with climatic characteristics was written.

  1. Section 2.2: There is almost no information on the radar data set and the datasets are not cited. Where is the data available? What is the scan strategy of the radars? How frequently are scan volumes provided? What is the elevation and azimuth angle spacing? What is the spacial and temporal resolution of the data? What is meant with "Beam angle" in Table 1? Which variables are used? Etc.

Response:Information in sections 2.1 and 2.2 is presented in more details.

  1. There is even less information on the lightning data. It needs to be described in detail.

Response: More details about lightning detection systems and lightning data are presented in section 2.2

9.Line 159: What are "statistically significant volumes of data"?

Response: Sorry, this had come from typical Russian expression in scientific literature, means a lot of analyzed cases. It is changed on –the significant number

  1. What is Y called and what does the cr stand for in Ycr?

Response: As far as we know, this criteria is used mostly in Russia to forecast thunderclouds. The higher Y the more is probability of lightning occurrence.  There is some critical value of Y – (Ycr) with which Y is compared, and probability of lightning is determined. The value of Ycr was assessed experimentally and probability also. The method is used at routine meteorological observations. It is called Y criteria in Russia, but we think that it is better to call “Thundercloud criteria Y” in the case of International Journal.

  1. Lines 132-165: What data is used to identify the isotherms? It needs to be described in detail as well.

Response: A phrase is added to paragraph 2.3. - Nearest rawisonde sounding is used to determine a height of the isotherm.

  1. Line 166: It is unclear what is meant with thunderstorm occurrence. Do you mean that an existing storm will develop lightning activity, turning from a "regular" storm into a thunderstorm?

Response: Thunderstorm occurrence means the possibility of lightning appearance in the cloud with Z3 and top height H. Changes were made in the text (paragraph 2.3)

  1. Please provide references for equations 2-6. Please also clarify if Z, Zw, and Zi are in linear or logarithmic units and how they are different from the Z3 mentioned before.

Response: References for the equations 4-6 are in the text and for the equation 3 is added. We do not think that there is any need for reference for equation 2. Z, Zw, and Zi – all are in mm6/m3, added to the text. Yes Z3 is different, it is reflectivity at a height 2–2.5 km above the zero-degree isotherm.

  1. What are b and bw in equation 3? What are the bins mentioned in line 176?

Response:b- means bin;  bw- bin with water drops. It is regular formula for calculation of reflectivity.

  1. References for the "previous investigations" are missing at line 192.

Response: references are added.

  1. Starting line 197: The description of the clouds should be in the "Methods" section.

Response: done.

16.Line 204: What were the criteria for selecting the 60 clouds?

Response: The criteria was the transition to thundercloud stage, it is mentioned in the text - …convective clouds transitioning into the thunderclouds were selected for each region.

17.In the description of the studied cloud groups (line 109) only very few clouds are mentioned but now 60 clouds in each region are described. This is very confusing.

Response: Yes, you are right. We changed text in paragraph 2, trying to clarify the data which was used for this investigation

18.Line 205: Since we know nothing about the timing and time resolution of the lightning and the radar data it is unclear how the time "10 minutes before lightning onset" is determined. All of this information belongs in the "Methods" not the "Results" section.

Response:- We changed text in paragraph 2, trying to clarify the data which was used for this investigation

19.Line 207: 5 dBZ does not represent cloud top height.

Response:-We changed text and use term - maximum height of the 5 dBZ reflectivity H5dBZ

20 Line 208: How is the maximum reflectivity defined? Of the whole cloud? In a vertical column?

Response: It is accepted now that maximum reflectivity of the cloud is calculated for cloud volume, the same is in our investigation.

  1. Line 211: Since we don't even know if the radar data is in polar or Cartesian coordinates, or what their resolution is, how are we supposed to interpret "Radar data ... were horizontally and vertically averaged in 1 km steps."?

Response: Radar measurements are carried out in polar coordinates, practically all modern processing software transform this data into Cartesian coordinates. Usually, there is no need to mention this, except cases when polar data is used. In our case, the used information from all radars was in  Cartesian coordinates. We do not think that there is need to mention this in the article.

  1. Line 231: How is the "lightning frequency distribution" defined?

Response:Number of lightning strikes during a radar scan  were calculated. Then frequency of lightning per minute was calculated. All the data for the investigated clouds formed a distribution of lightning.

Reviewer 3 Report

The manuscript Investigation of Thundercloud Features in Different Regions compares the statistical properties of thunderstorms and their development in several regions of the world. The authors use different types of data and use basic statistical characteristics. The manuscript is extensive and not easy to read, both because of its very descriptive nature and because some formulations require refinement.

Correlation is used as one of the statistical tools, I assume Pearson correlation coefficient. The application of this correlation is very problematic, because the correlated quantities do not have the necessary properties. Therefore, a different correlation coefficient should be used or an explicit justification for using Pearson correlation should be mentioned.

I may have missed something, but I don't know, how you got the temperature you use.

I also have a comment on the structure of the article. I don't see any discussion in the "Discussion" section. Here you only briefly describe the main results from the "Results" section. Therefore, I propose to cancel this part and call the previous section "Results and discussion".

Captions of the figures should be more detailed - self-explanatory.

That is why I propose a "major revision".

Specific comments:

Line 162: “is” is missing

Line 180: I suppose that Dbi instead of Di should be.

Line 185: I do not understand the statement in brackets. Could you explain it?

Last paragraph on p. 6: Do I understand well that you use p-value as a measure of similarity? Could you say it clearly?

In several places you use zero-degree isotherm, melting layer, melting zone or bright band. Do you use is it as synonyms or not?

 Line 238-240: Could you say it more clearly?

Line 240-241: I understand that here and elsewhere you assume that high reflectivity indicates large particles, but here also supercooled water can play a role. Could you explain it in more details.

Line 295: Here and elsewhere what do you mean by “supercooled”?

Line 299: “are” should be removed.

Paragraph 309-313: What do you mean by radiation temperature? Is it brightness temperature? It is not true that e.g. 10.8 channel characterizes the intensity of cloud development (cirrus may have high values). In my opinion only dif4 is a relatively good characteristics of the intensity.

Line 360: Please reconsider formulation “crystal melting”.

Line 416-417: As far as I know interactions of large ice particles (graupel) and small ice particles are now considered as the main source of electrification. Could you explain your statement and add a reference.

Line 423: Do you mean precipitation on the ground?

Line 452: What do you mean “by modulus”?

Author Response

Reviewer #3:

1.The manuscript Investigation of Thundercloud Features in Different Regions compares the statistical properties of thunderstorms and their development in several regions of the world. The authors use different types of data and use basic statistical characteristics.

Response: We thank the reviewer very much for his/her review. The following insightful comments and suggestions are very helpful to improve our manuscript. Our point-by-point responses are listed below.

2.The manuscript is extensive and not easy to read, both because of its very descriptive nature and because some formulations require refinement.

Response: We agree, to clarify the main ideas of the manuscript - a contribution of the article to the investigated problem is formulated in the introduction. Chapter 2 is significantly changed; here we tried to clarify information about the used data and methods of analysis.

3.Correlation is used as one of the statistical tools, I assume Pearson correlation coefficient. The application of this correlation is very problematic, because the correlated quantities do not have the necessary properties. Therefore, a different correlation coefficient should be used or an explicit justification for using Pearson correlation should be mentioned.

Response: Thank you, distributions typically are not normal, hence Pearson correlation is not proper in this case. We have recalculated correlations and have used Spearman correlation method. It is noted in the text. There were some changes in the text in accordance with the new data.

  1. I may have missed something, but I don't know, how you got the temperature you use.

Response: We have  used rawisonde data to get temperature profile. We also have used radiation (brightness temperature) from Meteosat (SEVERI instrument) measurements. To clarify this we added a phrase in section 2.3 -   Nearest rawisonde sounding is used to determine a height of the isotherm.

5.I also have a comment on the structure of the article. I don't see any discussion in the "Discussion" section. Here you only briefly describe the main results from the "Results" section. Therefore, I propose to cancel this part and call the previous section "Results and discussion".

Response:We agree with a remark. Our original version of the paper included only one section "Results and discussion", but editorial board had insisted to have both sections.

  1. Captions of the figures should be more detailed - self-explanatory.

Response:We added information to captions.

  1. That is why I propose a "major revision".

Response: We have tried to clarify the paper.

 8 Line 162: “is” is missing

Response: Thank you. It is inserted.

  1. Line 180: I suppose that Dbi instead of Di should be.

Response: We decided to rewrite this part, fig 1 was changed and climatologic description was simplified.

  1. Line 185: I do not understand the statement in brackets. Could you explain it?

Response:The basiс radar equation shows that reflectivity depends on particle diameter in 6 power multiplied by the  refractive index. For ice particle the refractive index is proportional to ice density in second power. Ice crystals density depends on their sizes , hence approximations formulas 5-6 appear. More inf is in publication - ref 45.

11.Last paragraph on p. 6: Do I understand well that you use p-value as a measure of similarity? Could you say it clearly?

Response:We tried to clarify this, now the phrase is - The higher the p-value, the smaller the difference between the distributions of two samples, so this is  used to assess the similarity of the analyzed distributions.

  1. In several places you use zero-degree isotherm, melting layer, melting zone or bright band. Do you use is it as synonyms or not?

Response: Terms - melting layer, melting zone or bright band- are used usually in radar meteorology as synonyms. We use them, to make the text more literary. Zero-degree isotherm is used to indicate its height.

  1. Line 238-240: Could you say it more clearly?

Response:We have clarified  the text in paragraph 3.1 by excluding from discussion parameters, characterizing characteristics relative to zero isotherm.

  1. Line 240-241: I understand that here and elsewhere you assume that high reflectivity indicates large particles, but here also supercooled water can play a role. Could you explain it in more details.

Response: We use term large hydrometeor, which include both ice and liquid in this part of the text. Supercooled water is usually associated with  droplets. Large particles determine in great  extent the reflectivity, of course liquid precipitating drops have great impact on reflectivity alongside with large crystals. We try to be careful in interpretation of the results.  

  1. Line 295: Here and elsewhere what do you mean by “supercooled”?

Response: We follow the usual definition – liquid drops and droplets at temperature less than 0C.

  1. Line 299: “are” should be removed.

Response: Thank you, done.

  1. Paragraph 309-313: What do you mean by radiation temperature? Is it brightness temperature? It is not true that e.g. 10.8 channel characterizes the intensity of cloud development (cirrus may have high values). In my opinion only dif4 is a relatively good characteristics of the intensity.

Response: Radiation temperature is synonym to brightness temperature. In general, it is more used to characterize middle or far infrared range. You are right, 10.8 channel can be influenced and influenced by high level clouds. In our case, we studied a development of every cloud under consideration during its life cycle and have retrieved cloud top temperature at 10.8 channel. The dynamics of this value characterizes the living stage of the cloud. We tried to clarify the method in the revised version, and changed text in paragraph 2.

  1. Line 360: Please reconsider formulation “crystal melting”.

Response:  Heights of maximum reflectivity HZm were 3.5-5 km, the only exception is N-W_GR, where the height of max reflectivity was significantly less and near cloud base, indicating that it could be the result of large ice particles melting and producing a bright band.

  1. Line 416-417: As far as I know interactions of large ice particles (graupel) and small ice particles are now considered as the main source of electrification. Could you explain your statement and add a reference.

Response:  It was a mistake, the phrase is changed - The presence of large ice particles enable the interactions with cloud crystals that result in Cb electrification [6].

The presence of larger ice particles enable the interactions with cloud crystals that result in Cb electrification [6].

21.Line 423: Do you mean precipitation on the ground?

Response:  Radars measure precipitation higher than at the ground level. It is 1 km height as a rule. It is accepted that they are  equal to  that, measured at a ground level, though the difference exists due to their evaporation.

  1. Line 452: What do you mean “by modulus”?

 Response:  The modulus is absolute value.

Thanks for your time,

Authors

Round 2

Reviewer 2 Report

The authors have made an effort to improve the quality of the paper. Unfortunately, the main points of criticism still stand. The paper is lacking the basics for scientific publishing: adequate description of the data and methods. Without that it is impossible to judge the validity of the conclusions. The authors have had the opportunity to correct these shortcomings but decided to re-submit with minor revisions that were simply not enough. My recommendation is therefore still the same as last time: reject.

Major comments:

It is unclear if the conclusions drawn from the analysis are valid. The paper is based on statistical analyses but the sample sizes are so small, often single cases, that a statistical approach likely does not give meaningful results. Also, it is hard to tell if the results are statistically significant since it is often unclear which data exactly was used to calculate the statistical parameters as methods are not adequately described (see below).

The datasets used are still not adequately described. Some are not described at all and for others crucial information is missing. None of the datasets has a citation or information on where it can be obtained.

The methods are still not adequately described. Just two examples: Cloud tops from radar measurements and satellite data are compared but the exact details of the comparison are not provided. The Spearman correlation coefficient is calculated but it is not clear on what data exactly. There are many more examples like these.

Uncertainties in the analysis are not addressed sufficiently. The authors mention that e.g. differences in sensors could be a significant error source but do not quantify or at least give an estimate on the magnitude of these errors.

The paper is not well written. The organization is chaotic and terminology inconsistent and not clear. There are many spelling and grammatical errors. Figures and tables are not referenced in the text and the discussion is therefore hard to follow.

Specific comments:

The paragraph starting line 91 belongs in the conclusions, not the introduction.

I appreciate the new Figure 1. However, it is not referenced in the text. It should frequently be mentioned in Sec. 2.1. Please also label the sub-panels and reference them accordingly in the figure caption and text.

Line 112: Maharashtra and Karnataka are mentioned in the text but Fig. 1 shows Aurangabad and Gadag. Please make the figure and the text consistent.

Line 124: I disagree with the statement that "Areas of study in India are in flat regions" The weather and climate in these areas are heavily influenced by the Western Ghats (which reach altitudes of >2500 m) which has been studied extensively.

Line 140: These statements don't make a lot of sense. "simple statistics" is not a "type of data".

Line 151: On which basis were the 60 convective clouds selected? What were the criteria for their selection?

Line 155: Again, "type of data" is not the right term. Do you mean "type of analysis"?

Line 190: Can you give an exact number instead of "Several tens"?

I appreciate the author's efforts to give more information on the radars used. Contrary to the author's statement that "there is no need to mention" that the radar data is in Cartesian coordinates I believe it is essential to mention this fact and even more essential to mention what the resolution of the Cartesian grids are! Is the resolution the same for all radars used?

Line 217: LF and VHF have been used many times already and need to be defined much earlier, at their first use.

The radar and lightning data are still not cited properly. Where is the data available? Please add dataset citations.

Line 249: The rawinsonde data needs to be described and cited in the data section.

Line 289: As already mentioned in my last review: It is important to know what the data resolution was before the horizontal and vertical averaging!

Figure 2: As in Fig. 1, please add labels (a, b, c, d) to the subplots and reference them in the text. Also, please add the unit (min) to the x-axis. It would also be very helpful to add titles to the sub-panels.

Figure 2: Since the scan times of the radars have large variations and the largest is 7.5 minutes, how can you have data at exactly the time of the first lightning strike and exactly 10 minutes before the first lightning strike? How would that work for a 7.5 minute scan time? This is why it is so important to give both spacial and temporal resolution of both the radar and the lightning data!

Paragraph starting line 323: This paragraph is rather confusing as several processes are mixed together. First it is stated that an increase in reflectivity is caused by an increase in particle size. Later it is stated that the large reflectivity at the -10 min stage is due to the melting layer. So are you saying that the particle size in the melting layer increases? Also, the fact that high reflectivity observations in northwestern Russia are attributed to a melting layer but in other regions are not, may imply that the storms in nw Russia may be in a later, more stratiform stage of development because the melting layer is usually not easily identified in updraft regions of early convective stages. Since one would expect lightning more in the updraft regions it would be good to get an explanation for why first lightning is detected in storms with an already developed melting layer.

Line 350: It is unclear how the periods with VHF and LF lightning strikes were chosen. Are they distinct time periods or do they overlap in time? How many clouds were in the "period of radar scans when LF lightning discharges were observed"? Was it a statistically significant sample? These periods need to be clearly defined with all their properties (how many, how long, etc.).

Line 353: What are the "above-mentioned categories"? Are those what you referred to as "Groups" earlier?

Figures 3 and 4: Again, please label the sub-plots and refer to them in the text. Add titles to the sub-plots.

Line 386: How do you know that the volume is supercooled? From reflectivity alone it is not possible to distinguish between supercooled liquid and ice phase.

Starting line 395: The satellite data has not been mentioned. Every single dataset used and how it was processed needs to be described in detail in the data section!

Line 413: How much altitude difference was between the height of the maximum reflectivity and the zero degree isotherm? Also, why don't you check in the data if the maximum reflectivity was part of a bright band? Since all the necessary data is available, why speculate?

Line 460: How do you know that crystal melting took place and that it was a bright band. Did you check?

Line 463: Again, how do you know that the volume is supercooled? Your terminology is contradictory as you are speaking of large ice hydrometeors within the supercooled volume. Also, why would the existence of large ice hydrometeors lead to a large supercooled volume?

Line 468: Please provide a reference for that statement.

Once the difference between the LF and VHF periods has been explained (see comment above) it would be good to compare these two groups of periods with each other and explain the differences and similarities.

Line 476 and throughout the rest of the manuscript: Again, I don't think you are using the term "supercooled" correctly.

Section 3.4: References to Table 2 are missing.

Beginning of Section 3.4: How much data went into the calculation of the correlation coefficients? Are the results statistically significant?

Beginning of Section 3.4: There are now so many different cases lumped together in different ways that it is extremely hard to tell them apart and to know which correlation coefficients are calculated for which cases. Some are listed in Table 2 but some are only mentioned in the text. Since the text doesn't reference the table it is impossible to follow the discussion. This part of the paper needs to be re-written with consistent terminology.

Line 490: Please explain what A and B are. The reader can guess but it should be in the text.

Figure 5: It is unclear what data is plotted in this figure and what the uncertainties are. It seems like the data the linear approximations come from are not plotted? Again, how much data was there?

Equation 7: Since no uncertainty is given it is completely unclear if this equation is of any value.

Section 3.6: This is not my area of expertise so I cannot comment on this section.

Line 616: How does the current study confirm "differences between ... land and oceanic weather systems" if no oceanic weather systems were studied?

Starting line 629: There is no information in the data that tells whether the dV35 was supercooled liquid or ice phase. Please do not make statements about the influence of supercooled liquid if you don't know that that is what you are observing.

Sections 4 and 5: The discussion and conclusion sections are basically identical and should be combined into one section.

Author Response

Reviewer #2:

The authors have made an effort to improve the quality of the paper. Unfortunately, the main points of criticism still stand. The paper is lacking the basics for scientific publishing: adequate description of the data and methods. Without that it is impossible to judge the validity of the conclusions. The authors have had the opportunity to correct these shortcomings but decided to re-submit with minor revisions that were simply not enough. My recommendation is therefore still the same as last time: reject.

Response: We thank the reviewer very much for his/her review. We thank the reviewer for his/her time and for valuable comments and especially new ideas. Our point-by-point responses are listed below.

Major comments:

  1. It is unclear if the conclusions drawn from the analysis are valid. The paper is based on statistical analyses but the sample sizes are so small, often single cases, that a statistical approach likely does not give meaningful results. Also, it is hard to tell if the results are statistically significant since it is often unclear which data exactly was used to calculate the statistical parameters as methods are not adequately described (see below).

Response: Information about samples is presented in section 2. Sample sizes are different for the regions under consideration and usually equals to several tens of radar scans. Detailed information about the used data is also presented in our earlier publications referred in this section.

  1. The datasets used are still not adequately described. Some are not described at all and for others crucial information is missing. None of the datasets has a citation or information on where it can be obtained.

Response. No information from datasets is available online but detailed data with scientific analysis is presented in our earlier articles, mentioned in the text. We are not planning to form a dataset but are ready to present any of the data if there will be interest in it.

3.The methods are still not adequately described. Just two examples: Cloud tops from radar measurements and satellite data are compared but the exact details of the comparison are not provided. The Spearman correlation coefficient is calculated but it is not clear on what data exactly. There are many more examples like these.

Response. Our earlier publications present detailed information about the differences of cloud top measurements using satellite and radar data. It was shown that in many cases radar measurements overestimate cloud top height. It is not the goal of the current article to discuss this issue. Spearman correlation was used to get relations between lightning frequency and radar parameters, it is mentioned in section 3.4.

4.Uncertainties in the analysis are not addressed sufficiently. The authors mention that e.g. differences in sensors could be a significant error source but do not quantify or at least give an estimate on the magnitude of these errors.

Response. Main sources of information used for current investigation are: radar data, lightning detection data and satellite data. We agree with a reviewer that there can be differences between sensors. For example, radars can have different errors, especially different types of radars. This problem is not the issue of the article and we cannot solve so significant problems within the frames of the current manuscript. The same concerns lightning and satellite measurement and also rawisonde data. We have given rather brief  information about instruments  in section 2.2 and references to the readers to get more details.

5.The paper is not well written. The organization is chaotic and terminology inconsistent and not clear. There are many spelling and grammatical errors. Figures and tables are not referenced in the text and the discussion is therefore hard to follow.

Response. The organization of the text seems to be organized. Introduction presents literature review and main ideas of the investigation. It followed by description of the regions, instruments, methods   and data. The following chapter analysis characteristics of the clouds during first discharge and so on. We do not see any chaos. Spelling and grammar were checked. The final version was checked by the second author whose language is native English. We have checked the missed references on figures and tables and made corrections.

Specific comments:

6.The paragraph starting line 91 belongs in the conclusions, not the introduction.

Response. One of the reviewers recommended to formulate main results at the end of introduction. We agreed and introduced one very short paragraph (5 phrases) into the end of introduction. We think that it is convenient for readers.

  1. I appreciate the new Figure 1. However, it is not referenced in the text. It should frequently be mentioned in Sec. 2.1. Please also label the sub-panels and reference them accordingly in the figure caption and text.

Response. Corrections were made.

  1. Line 112: Maharashtra and Karnataka are mentioned in the text but Fig. 1 shows Aurangabad and Gadag. Please make the figure and the text consistent.

Response. It is mentioned that cloud characteristics were gotten near these cities and they are located in the consequent provinces (line 186). We think that the following mentioning of the provinces is more reasonable as these are small cities not well known to readers.

  1. Line 124: I disagree with the statement that "Areas of study in India are in flat regions" The weather and climate in these areas are heavily influenced by the Western Ghats (which reach altitudes of >2500 m) which has been studied extensively.

Response. The phrase was corrected.

  1. Line 140: These statements don't make a lot of sense. "simple statistics" is not a "type of data".

Response. Changes were made.

  1. Line 151: On which basis were the 60 convective clouds selected? What were the criteria for their selection?

Response. This was the simple criteria – transitioning to thundercloud stage.

  1. Line 155: Again, "type of data" is not the right term. Do you mean "type of analysis"?

Response. Here, it is the right term, data was retrieved from the previously analyzed case studies.

  1. Line 190: Can you give an exact number instead of "Several tens"?

Response. Done.

  1. I appreciate the author's efforts to give more information on the radars used. Contrary to the author's statement that "there is no need to mention" that the radar data is in Cartesian coordinates I believe it is essential to mention this fact and even more essential to mention what the resolution of the Cartesian grids are! Is the resolution the same for all radars used?

Response. The resolution of the radars differ, but in most cases it is near 500m,  and it is determined by the radar technical characteristics  and programs of radar data processing. We agree that it would be better to have data from fully identical systems installed in the regions under consideration, but there is no such data as far as we know.  

  1. Line 217: LF and VHF have been used many times already and need to be defined much earlier, at their first use.

Response. Done.

  1. The radar and lightning data are still not cited properly. Where is the data available? Please add dataset citations.

No information from datasets is available online but detailed data with scientific analysis is presented in our earlier articles, mentioned in the text. We are not planning to form a dataset but are ready to present any of the data if there will be interest in it.

17 Line 249: The rawinsonde data needs to be described and cited in the data section.

Response. Rawisonde data is a usual data, used for meteorological analysis, we do not think that there is need in any details. It was mentioned that nearest station data is used in this analysis.

18 Line 289: As already mentioned in my last review: It is important to know what the data resolution was before the horizontal and vertical averaging!

Response. Programs, which are used for radar data processing,  usually form bins of equal sizes in horizontal and vertical direction. Details and titles of such programs are mentioned in the referred literature, where cases, which form the base of this investigation, are presented in details. 

19 Figure 2: As in Fig. 1, please add labels (a, b, c, d) to the subplots and reference them in the text. Also, please add the unit (min) to the x-axis. It would also be very helpful to add titles to the sub-panels.

Response. Done.

20 Figure 2: Since the scan times of the radars have large variations and the largest is 7.5 minutes, how can you have data at exactly the time of the first lightning strike and exactly 10 minutes before the first lightning strike? How would that work for a 7.5 minute scan time? This is why it is so important to give both spacial and temporal resolution of both the radar and the lightning data!

Response. We clarified the method to get lightning frequency by adding the following “All lightning flashes, which were detected in a cloud  during  radar scan, were counted, and  averaged frequency for  scan  in min-1 was determined. This frequency was used for further data processing.” A scan with lightning was used in this part of work and the previous one. You are right, they can differ in their duration. We changed the text in paragraph 2 to clarify the method of getting data.

  1. Paragraph starting line 323: This paragraph is rather confusing as several processes are mixed together. First it is stated that an increase in reflectivity is caused by an increase in particle size. Later it is stated that the large reflectivity at the -10 min stage is due to the melting layer. So are you saying that the particle size in the melting layer increases? Also, the fact that high reflectivity observations in northwestern Russia are attributed to a melting layer but in other regions are not, may imply that the storms in nw Russia may be in a later, more stratiform stage of development because the melting layer is usually not easily identified in updraft regions of early convective stages. Since one would expect lightning more in the updraft regions it would be good to get an explanation for why first lightning is detected in storms with an already developed melting layer.

Response. Melting layer is located at relatively low altitudes in n-w clouds. Large ice particles in the form of precipitation start to melt there. So, an increase in reflectivity is explained first by the presence of large particles which is emphasized by melting. We did not observed such increase in Indian clouds, it can be due to limited amount of large ice particles.

  1. Line 350: It is unclear how the periods with VHF and LF lightning strikes were chosen. Are they distinct time periods or do they overlap in time? How many clouds were in the "period of radar scans when LF lightning discharges were observed"? Was it a statistically significant sample? These periods need to be clearly defined with all their properties (how many, how long, etc.).

Response. It is written in the text: “These characteristics were chosen for clouds at the  period of radar scans when LF lightning discharges (further LF lightning) were observed  in these clouds just during the scans. “ So, only scans when LF discharges were observed are under consideration here. A next paragraph deals with VHF discharges. During some scans both were observed, in other cases only one of them. Information about the number of radar scans under consideration is inserted to the text.

  1. Line 353: What are the "above-mentioned categories"? Are those what you referred to as "Groups" earlier?

Response. Yes, categories are changed to groups.

  1. Figures 3 and 4: Again, please label the sub-plots and refer to them in the text. Add titles to the sub-plots.

Response. Done.

  1. Line 386: How do you know that the volume is supercooled? From reflectivity alone it is not possible to distinguish between supercooled liquid and ice phase.

Response. A definition of dV35 is given at the beginning of the paragraph 3.1. We do not try to distinguish water from ice in current work.

  1. Starting line 395: The satellite data has not been mentioned. Every single dataset used and how it was processed needs to be described in detail in the data section!

Response.Satellite data was available only for limited number of cases under consideration. Main results, which were gotten with satellite measurements are presented in paragraph 3.2. We referred our earlier publications where details are presented.

  1. Line 413: How much altitude difference was between the height of the maximum reflectivity and the zero degree isotherm? Also, why don't you check in the data if the maximum reflectivity was part of a bright band? Since all the necessary data is available, why speculate?

Response. We did not investigate a position of maximum reflectivity to Zero degree isotherm. May be it is a good idea for our future investigations.

  1. Line 460: How do you know that crystal melting took place and that it was a bright band. Did you check?

Response. Bright band is observed in clouds near isotherm 0 degree due to crystal melting. It is an accepted fact in radar meteorology, so we only state this for our case.

  1. Line 463: Again, how do you know that the volume is supercooled? Your terminology is contradictory as you are speaking of large ice hydrometeors within the supercooled volume. Also, why would the existence of large ice hydrometeors lead to a large supercooled volume?

Response. Supercooled volume definition is given at the beginning of the paragraph 3.2. This volume includes crystal particles and supercooled drops in many cases, though it can be only crystals in accordance with the definition. Large hydrometeors determine in great extent reflectivity in accordance with radar equation.

  1. Line 468: Please provide a reference for that statement.

Response. Done.

31.Once the difference between the LF and VHF periods has been explained (see comment above) it would be good to compare these two groups of periods with each other and explain the differences and similarities.

Response. It has not been done but an idea is interesting. Probably, we’ll try to check the possible difference in future investigations.

  1. Line 476 and throughout the rest of the manuscript: Again, I don't think you are using the term "supercooled" correctly.

Response. This is simply definition, indicating that there can be supercooled drops in this volume.

  1. Section 3.4: References to Table 2 are missing.

Response. There is a reference at line 490.

  1. Beginning of Section 3.4: How much data went into the calculation of the correlation coefficients? Are the results statistically significant?

Response. Correlation coefficients were calculated for each case study referred in section 2. Only significant correlation coefficients were under consideration further.

  1. Beginning of Section 3.4: There are now so many different cases lumped together in different ways that it is extremely hard to tell them apart and to know which correlation coefficients are calculated for which cases. Some are listed in Table 2 but some are only mentioned in the text. Since the text doesn't reference the table it is impossible to follow the discussion. This part of the paper needs to be re-written with consistent terminology.

Response. It is mentioned at the beginning of the paragraph that case studies are under consideration here. We do not mention the previous definition –Groups-. To clarify the used data we have added references in table 2.

  1. Line 490: Please explain what A and B are. The reader can guess but it should be in the text.

Response. These are ”coefficients for linear relations between fLF and dV35”, it is just mentioned in line 490.

  1. Figure 5: It is unclear what data is plotted in this figure and what the uncertainties are. It seems like the data the linear approximations come from are not plotted? Again, how much data was there?

Response. Fig. 5 is a linear approximation between the studied parameters for different cases. Correlation coefficients were obtained for all scans with lightning for these cases. A number of scans is different for every cloud which were studied. The main idea to present this plot is to show that the needed volume to form one lightning strike does not differ much between different clouds.

  1. Equation 7: Since no uncertainty is given it is completely unclear if this equation is of any value.

Response. This equation can be used only for assessments; it is mentioned in the text.

  1. Section 3.6: This is not my area of expertise so I cannot comment on this section.
  2. Line 616: How does the current study confirm "differences between ... land and oceanic weather systems" if no oceanic weather systems were studied?

Response. We agree, the phrase is excluded.

41.Starting line 629: There is no information in the data that tells whether the dV35 was supercooled liquid or ice phase. Please do not make statements about the influence of supercooled liquid if you don't know that that is what you are observing.

Response. This is simply definition, indicating that there can be supercooled drops in this volume.

42.Sections 4 and 5: The discussion and conclusion sections are basically identical and should be combined into one section.

Response. Original text had a combined paragraph, but it was changed for two due to Editorial Board requirement.

Reviewer 3 Report

I'm on vacation now and I have limited time options. However, I briefly went through the text and I think the bugs have been fixed. 
From a professional point of view, I recommend accepting the article.

Author Response

Reviewer #3:

I'm on vacation now and I have limited time options. However, I briefly went through the text and I think the bugs have been fixed. 
From a professional point of view, I recommend accepting the article.

Response: We thank the reviewer very much for his/her time. Your recommendations have been very useful. We tried our best to improve the text.

Sincerely yours,

Authors

This manuscript is a resubmission of an earlier submission. The following is a list of the peer review reports and author responses from that submission.